# FLOWGEN: SYNTHESIZING DIVERSE FLOWCHARTS TO ENHANCE AND BENCHMARK MLLM REASONING

**Kaiwen Shi, Sichen Liu, Ziyue Lin, Hangrui Guo, Gong Cheng**[*]
State Key Laboratory for Novel Software Technology, Nanjing University, Nanjing, China
{kaiwenshi, sichenliu, 221240094, 221240069}@smail.nju.edu.cn
gcheng@nju.edu.cn

## ABSTRACT

Flowcharts are widely used to represent processes and relationships through intuitive visual representations. However, accurately interpreting these diagrams remains challenging due to their structural complexity and high visual diversity. Existing flowchart datasets often lack fine-grained control over key properties such as graph complexity and rendering style, limiting their utility for training and testing of multimodal large language models (MLLMs) on visual reasoning tasks. To address these limitations, we introduce FlowGen, a controllable synthesizer that generates flowcharts that have customizable structural features and supports multiple renderer backends. FlowGen enables fine-grained control over graph properties such as graph order and size, branched arrows, and nested subgraphs, facilitating systematic evaluation of MLLMs' capabilities. Extensive experiments on open-source and proprietary MLLMs show that training on FlowGen substantially improves flowchart parsing and question answering (QA), while also enhancing generalization to other public datasets. Furthermore, FlowGen provides challenging test datasets that expose consistent weaknesses in current MLLMs, particularly related to high structural complexity and varied rendering styles. Our code and data are publicly available at https://github.com/nju-websoft/FlowGen.

## 1 INTRODUCTION

Flowcharts, as illustrated in Figure 1, are widely used to represent processes, relationships, and workflows, prevalent in domains such as scientific communication, business process modeling (BPMN), software engineering documentation, and education (Bhushan et al., 2024). As a graphical language, flowcharts integrate symbolic information with spatial layout, enabling humans to quickly understand complex processes. However, this same structural complexity makes automatic parsing highly challenging (Montalvo, 1990). Building models to extract structured representations from flowcharts is crucial for enabling downstream applications, including code generation, knowledge extraction from documents, automated reasoning over workflows, and multimodal question answering (Vasudevan et al., 2008; Bashir & Giri, 2013; Hu et al., 2024).

**Motivation.** Recent advances in vision language models (VLMs) and MLLMs have shown potential to understand flowcharts, enabling end-to-end reasoning from images (Ye et al., 2024). However, the development of such capabilities is hindered by the limitations of existing datasets. On the one hand, existing *test sets* provide limited control over complex structures, such as branching factor and hierarchical nesting. They are also usually

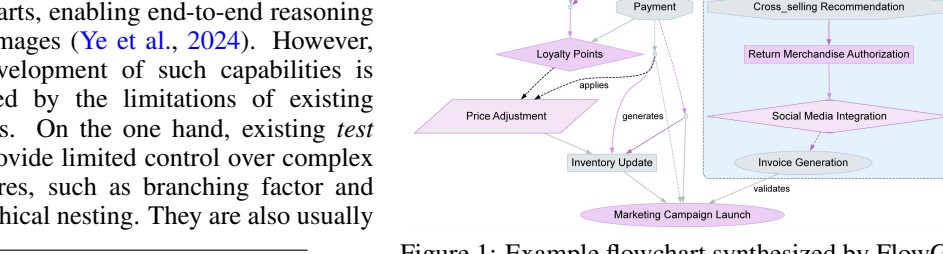

Figure 1: Example flowchart synthesized by FlowGen.

[*]Corresponding Author: Gong Cheng.

Table 1: Comparison between our proposed synthesizer (FlowGen) and existing flowchart datasets.

| Dataset | Complexity | Renderer(s) | Source | #Train | #Test |
|---|---|---|---|---|---|
| FlowchartQA (Tannert et al., 2023) | NA | Graphviz | Synthetic | Not released | Not released |
| FlowVQA (Singh et al., 2024) | NA | Mermaid | Synthetic | 1,319 | 953 |
| CBD (Bhushan & Lee, 2022) | NA | NA | Web crawling | 300 | 96 |
| FC_A (Awal et al., 2011) | Branching | NA | Hand-drawn flowcharts | 248 | 145 |
| FC_B (Bresler et al., 2016) | Branching | NA | Hand-drawn flowcharts | 280 | 196 |
| hdBPMN (Schäfer et al., 2021b) | Branching & Nesting | NA | Hand-drawn BPMN diagrams | 0 | 173 |
| FlowLearn (Pan et al., 2024) | Branching | Mermaid | Synthetic | 8,000 | 2,000 |
| **FlowGen (ours)** | **Branching & Nesting** | **Mermaid, Graphviz, PlantUML, Diagrams** | **Synthetic** | **Unlimited** | **Unlimited** |

rendered in a single visual style, which prevents evaluating the robustness to stylistic variation, while real-world flowcharts are highly diverse (Awal et al., 2011; Bresler et al., 2016; Schäfer et al., 2021b; Bhushan & Lee, 2022; Tannert et al., 2023). On the other hand, existing *training sets* are small, narrowly focused on specific domains, or missing important structural features (Sun et al., 2022; Singh et al., 2024; Bhushan et al., 2024; Pan et al., 2024). These issues hinder effective model training, and models fail when confronted with high structural complexity or cross-renderer settings. These limitations call for a more principled way to generate both training and test resources. More related work is discussed in Appendix B.

**Our Work.** To bridge both training and test gaps in previous work, as compared in Table 1, we introduce FlowGen, a controllable flowchart synthesizer that generates diagrams with tunable structural features and supports multiple rendering styles. Figure 1 presents an example flowchart synthesized by FlowGen, demonstrating split arrows, merge arrows, nested structures, and variations in colors, node shapes, and edge styles. This diversity enriches the distribution of generated flowcharts to better approximate real-world practices.

With FlowGen, we: (1) build a large-scale synthetic *training set* and demonstrate that fine-tuning open-source MLLMs on it substantially boosts their ability of flowchart parsing and flowchart QA, with strong transfer gains to multiple existing datasets, in some cases even surpassing proprietary MLLMs; (2) construct synthetic *test sets* with controlled complexity and renderer diversity, and use them to comprehensively test both open-source and proprietary MLLMs, revealing their consistent weaknesses in handling nested structures and cross-renderer generalization. Together, these results establish controllable flowchart synthesis as a practical and effective strategy to advance the machine understanding of such diagrams.

Our contributions are summarized as follows:

- We introduce a controllable flowchart synthesizer capable of generating flowcharts with adjustable structural complexity and multiple renderer styles, enabling the principled construction of both training and test resources.

- We synthesize large training sets and show that fine-tuning with them significantly improves MLLM reasoning on flowcharts, with strong transferability to a wide range of existing flowchart parsing and QA datasets.

- We synthesize challenging test sets that systematically expose weaknesses of both open-source and proprietary MLLMs, providing insights into their limitations under high complexity and cross-renderer variation.

## 2 SYNTHESIZER FRAMEWORK

Figure 2 provides an overview of our FlowGen synthesizer pipeline. The synthesizer converts a compact, user-specified configuration into structurally valid graphs, then semantically annotates and renders them with multiple, stylistically diverse backends. The pipeline is organized into three core stages: (1) configuration, (2) graph construction, and (3) rendering. These stages jointly enable fine-grained control over structural complexity, semantic richness, and visual appearance.

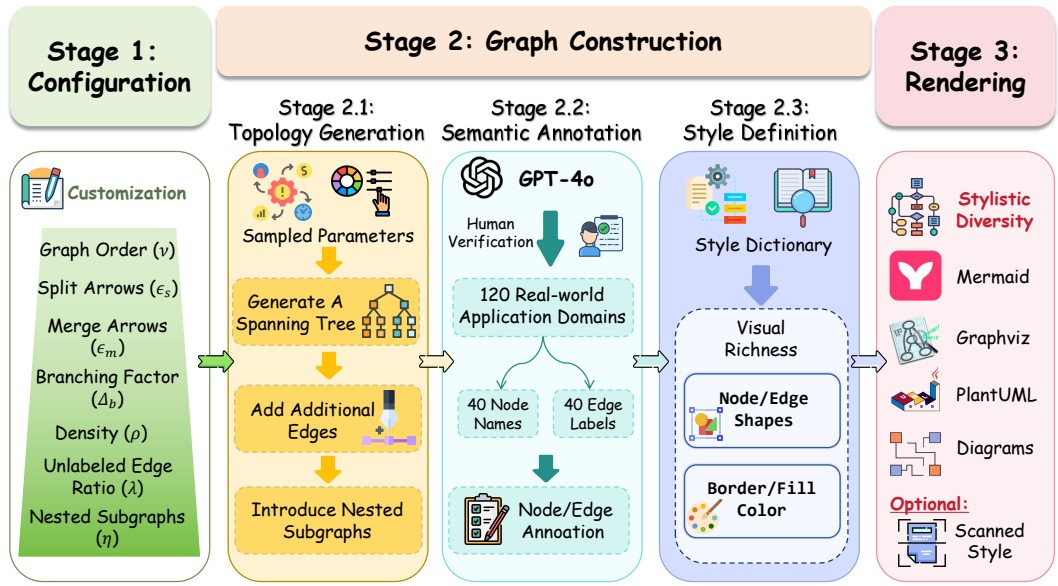

Figure 2: Overview of our FlowGen synthesizer pipeline.

## 2.1 CONFIGURATION

The synthesizer begins by sampling a set of structural parameters that govern the topological and semantic complexity of generated flowcharts. We define seven controllable parameters.

**Graph Order ($\nu$).** This specifies the total number of nodes in a flowchart, including those contained in nested subgraphs. We categorize the scale into three levels: *small* (8–12 nodes), *medium* (13–20 nodes), and *large* (21–30 nodes). This parameter is the primary factor controlling the overall length and complexity of the synthesized flowcharts.

**Split Arrows ($\epsilon_s$).** Split arrows introduce virtual nodes that model the divergence of a single process into multiple concurrent branches. Although they do not carry domain-specific semantics, they substantially increase structural variability. The parameter $\epsilon_s$ specifies the number of such virtual nodes, which regulates the branching complexity of the graph structure.

**Merge Arrows ($\epsilon_m$).** Merge arrows capture the convergence of multiple parallel streams into a single unified process. Each merging structure introduces a virtual node with multiple incoming edges, and the total number of such virtual nodes is controlled by $\epsilon_m$. Together with split arrows, these constructs enable simulation of complex control-flow patterns.

**Branching Factor ($\Delta_b$).** This governs the maximum fan-in and fan-out degrees of virtual nodes created for the split and merge arrows. During construction, up to $\Delta_b$ outgoing edges are assigned to a split node, and up to $\Delta_b$ incoming edges to a merge node. This provides fine-grained control over the degree of divergence and convergence.

**Density ($\rho$).** This is defined as the edge-to-node ratio $\rho = \epsilon/\nu$, where $\epsilon$ denotes the total number of edges and $\nu$ the number of nodes. This parameter controls the overall visual density of the graph. Higher values of $\rho$ indicate denser connectivity patterns, leading to visually cluttered graphs, whereas lower values produce sparser and more tree-like structures.

**Unlabeled Edge Ratio ($\lambda$).** This parameter $\lambda \in [0, 1]$ specifies the proportion of edges that do not carry textual labels. A higher $\lambda$ increases semantic ambiguity and makes the reasoning task

harder. When $\lambda = 0$, all edges are annotated with labels; when $\lambda = 1$, no edges contain labels. This parameter allows for controlled adjustment of semantic sparsity.

**Nested Subgraphs** ($\eta$). This determines the number of hierarchically embedded subgraphs, which introduce structural depth in addition to the global node count $\nu$. Each nested subgraph is assigned depth 1, and the edges incident to a subgraph connect directly to its internal nodes.

## 2.2 GRAPH CONSTRUCTION

Given the sampled parameters, the synthesizer constructs an abstract directed acyclic graph with semantically coherent labels to serve as the structural backbone of the flowchart. The process consists of three stages: topology generation, semantic annotation, and style definition.

### 2.2.1 TOPOLOGY GENERATION

We begin by generating a random spanning tree on $\nu$ nodes to establish an acyclic backbone for node positioning. To increase connectivity, additional edges are then introduced according to the sampled $\epsilon_s$, $\epsilon_m$, and $\Delta_b$. If the resulting number of edges is still below the target density $\rho$, further random acyclic edges are added until the specified ratio is met. To incorporate hierarchical depth, nested subgraphs are created by randomly selecting 2 to 5 connected nodes and grouping them into a higher-level node, controlled by the nesting count $\eta$. Nested subgraphs are constrained to be disjoint, preventing overlap and preserving clarity. If $\eta$ exceeds the feasible number of disjoint groupings for a given graph size, the generator halts further nesting to maintain structural validity.

### 2.2.2 SEMANTIC ANNOTATION

Once the abstract topology is constructed, the synthesizer assigns a semantic domain by sampling a topic from 120 predefined application domains. Each topic provides 40 node names and 40 edge labels, initially generated with GPT-4o and refined through human verification. The nodes and edges are annotated with topic-specific labels to maintain semantic coherence, and the edges are labeled with probability $1 - \lambda$, where $\lambda \in [0, 1]$ denotes the unlabeled edge ratio. Additional details of the application domains and representative node and edge vocabularies are provided in Appendix C.

### 2.2.3 STYLE DEFINITION

After topology generation and semantic annotation, the visual appearance of nodes and edges is determined using renderer-specific definitions from a predefined style dictionary. The node and edge shapes are randomly sampled from the available ones, with possible repetition to emulate the variability of real-world diagrams. To introduce color variation, five palettes are randomly drawn from a pool of 90 predefined schemes for each flowchart topic instance, and node borders, fills, and edge colors are sampled from these palettes. This rendering strategy enhances stylistic diversity and visual richness while ensuring thematic consistency within each flowchart synthesized.

## 2.3 RENDERING

The final stage of the synthesizer converts abstract annotated graph codes into visual flowcharts. To capture the diversity of real-world styles, we integrate four complementary rendering backends: Mermaid[1], Graphviz[2], PlantUML[3], and Diagrams[4]. Each backend provides its own syntax, layout algorithms, and visual conventions. For instance, Mermaid emphasizes lightweight web-native diagrams, Graphviz offers fine-grained control over hierarchical layouts, PlantUML supports modular subgraph definitions, and Diagrams provides programmatic composition of clusters and entities.

During rendering, abstract nodes and edges, which have been annotated with textual labels and style attributes, are translated into renderer-specific elements such as shapes, connectors, and nesting sub-

---

[1] https://mermaid-js.github.io/
[2] https://graphviz.org/
[3] https://plantuml.com/
[4] https://diagrams.mingrammer.com/

graphs. By applying the user-specified renderer, our synthesizer produces visually rich and diverse flowcharts that closely resemble diagrams created in real-world scenarios.

# 3 FLOWGEN FOR MLLMS TRAINING

In this section, we systematically evaluate the performance gains of MLLMs fine-tuned on the FlowGen-synthesized training set for flowchart parsing and flowchart QA tasks. We further conduct ablation experiments to dissect the impact of key synthesis components.

## 3.1 EXPERIMENTAL SETUP

### 3.1.1 EVALUATION DATASETS

We evaluate a wide range of MLLMs on datasets covering two task categories. For **flowchart parsing**, we use FlowVQA (Singh et al., 2024), CBD (Bhushan & Lee, 2022), FC_A (Awal et al., 2011), FC_B (Bresler et al., 2016), hdBPMN (Schäfer et al., 2021b), and FlowLearn (Pan et al., 2024). These datasets provide flowcharts with structural annotations, enabling direct assessment of parsing accuracy. For **flowchart QA**, we adopt FlowVQA, FlowLearn, AI2D (Kembhavi et al., 2016), and MISS-QA (Zhao et al., 2025), which contain diagram-based questions that require reasoning over both structure and semantics. Details of these datasets are provided in Appendix D.1.

### 3.1.2 EVALUATION METRICS

For flowchart parsing, we represent the graph structure as triplets that capture both topology and semantics. We distinguish: (1) labeled edges $(A, X, B)$, where $X$ denotes a directed relation from node $A$ to node $B$; (2) unlabeled edges $(A, connectedTo, B)$, which use the predefined relation $connectedTo$ to indicate an unlabeled relation between $A$ and $B$; and (3) nested subgraphs $(A, partOf, B)$, where node $A$ belongs to a container group $B$, capturing hierarchical containment.

**Strict Precision, Recall, and F1.** Based on triplets, we compute standard parsing metrics:

$$\text{Precision} = \frac{|T_{pred} \cap T_{gold}|}{|T_{pred}|}, \ \text{Recall} = \frac{|T_{pred} \cap T_{gold}|}{|T_{gold}|}, \ \text{F1} = \frac{2 \cdot \text{Precision} \cdot \text{Recall}}{\text{Precision} + \text{Recall}}. \quad (1)$$

These strict metrics require exact string matches between predicted ($T_{pred}$) and gold-standard triplets ($T_{gold}$), thus capturing topological fidelity but sensitive to textual mismatches.

**Relaxed Precision, Recall, and F1.** To tolerate minor textual variations, we define a relaxed matching rule based on the normalized Levenshtein distance (Levenshtein, 1965). The edit similarity (ES) between two strings $S_1$ and $S_2$ is given by:

$$\text{ES}(S_1, S_2) = 1 - \frac{\text{EditDistance}(S_1, S_2)}{\max(|S_1|, |S_2|)}. \quad (2)$$

A predicted triplet $(h_p, r_p, t_p)$ matches a gold-standard triplet $(h_g, r_g, t_g)$ if and only if

$$\min\{\text{ES}(h_p, h_g), \ \text{ES}(r_p, r_g), \ \text{ES}(t_p, t_g)\} > 0.85. \quad (3)$$

Using this relaxed criterion, we recompute precision, recall, and F1, which provide robustness to minor OCR errors while still strictly evaluating structural correctness.

**Accuracy.** For flowchart QA datasets (e.g., AI2D, MISS-QA), we report *Accuracy*, defined as the proportion of questions for which the model prediction exactly matches the gold-standard answer. This metric captures end-to-end reasoning performance and complements the triplet-based metrics.

### 3.1.3 PARTICIPATING MODELS

We evaluate the performance of 10 MLLMs across two categories: (1) **open-source models**, including Qwen2.5-VL-3B/7B (Bai et al., 2025), InternVL3-2B (Zhu et al., 2025), MiniCPM-V2.6-8B (Yao et al., 2024), Llava-V1.6-Mistral-7B-HF (Liu et al., 2024), and the Gemma3 series (4B,

Table 2: Configurations for synthesizing the training set. Ranges indicate values to sample.

| Difficulty | $\nu$ | $\epsilon_s$ | $\epsilon_m$ | $\Delta_b$ | $\eta$ | $\rho$ | $\lambda$ |
|---|---|---|---|---|---|---|---|
| Easy | (8, 12) | (0, 0) | (0, 0) | (1, 2) | 0 | (1.0, 1.1) | (0.3, 0.6) |
| Medium | (13, 20) | (1, 2) | (1, 2) | (2, 3) | 1 | (1.1, 1.2) | (0.6, 0.8) |
| Hard | (21, 30) | (2, 3) | (2, 3) | (3, 4) | 2 | (1.2, 1.3) | (0.8, 1.0) |

Table 3: Performance comparison of MLLMs on flowchart parsing datasets. We report strict F1 and relaxed F1 (rF1) scores. We compare each model with its variant fine-tuned on FlowGen (+ SFT).

| Model | FlowVQA | | CBD | | FC_A | | FC_B | | hdBPMN | | FlowLearn | |
|---|---|---|---|---|---|---|---|---|---|---|---|---|
| | F1 | rF1 | F1 | rF1 | F1 | rF1 | F1 | rF1 | F1 | rF1 | F1 | rF1 |
| *Proprietary Multimodal Large Language Models* | | | | | | | | | | | | |
| GPT-4o | 88.2 | 89.1 | 64.5 | 67.1 | 41.9 | 44.3 | 52.6 | 55.3 | 26.5 | 35.8 | 53.9 | 58.6 |
| Gemini-2.5-Flash | 62.3 | 63.3 | 63.2 | 66.2 | 38.5 | 40.9 | 52.9 | 55.8 | 8.1 | 10.5 | 82.1 | 86.2 |
| GLM-4V-Plus | 74.0 | 74.5 | 55.6 | 58.7 | 33.0 | 34.0 | 46.8 | 49.3 | 8.6 | 11.3 | 31.6 | 40.8 |
| *Open-Source Multimodal Large Language Models* | | | | | | | | | | | | |
| Qwen2.5-VL-3B | 12.7 | 13.0 | 17.9 | 18.3 | 2.8 | 3.0 | 8.6 | 9.1 | 2.3 | 3.3 | 20.6 | 25.4 |
| + SFT | **51.3** | **52.9** | **49.8** | **52.0** | **22.4** | **23.0** | **34.6** | **37.2** | **8.4** | **11.3** | **41.6** | **55.5** |
| Qwen2.5-VL-7B | 59.4 | 59.7 | 49.1 | 51.2 | 25.8 | 27.2 | 32.0 | 33.2 | 16.6 | 20.6 | 43.2 | 48.3 |
| + SFT | **70.9** | **71.9** | **55.4** | **57.7** | **29.5** | **31.4** | **40.8** | **43.7** | **18.3** | **24.2** | **60.1** | **71.2** |
| InternVL3-2B | 15.7 | 17.0 | 18.4 | 19.5 | 4.6 | 5.7 | 8.3 | 9.1 | **3.2** | **4.3** | 21.8 | 32.6 |
| + SFT | **34.6** | **39.4** | **37.8** | **40.7** | **13.7** | **14.7** | **21.6** | **24.3** | 2.1 | 2.8 | 13.2 | 25.2 |
| MiniCPM-V2.6-8B | 17.6 | 18.5 | 20.4 | 20.9 | 7.5 | 8.7 | 8.8 | 10.2 | 3.4 | 5.3 | 9.1 | 12.7 |
| + SFT | **44.0** | **47.2** | **37.9** | **39.5** | **13.4** | **14.3** | **22.9** | **24.9** | **3.6** | **5.7** | **20.0** | **31.0** |
| Llava-V1.6-Mistral-7B-HF | 2.6 | 3.3 | 12.2 | 13.1 | **2.8** | **3.7** | 4.5 | 5.3 | 1.0 | 1.3 | 3.6 | 8.9 |
| + SFT | **4.9** | **8.7** | **14.0** | **17.4** | 2.3 | 2.7 | **4.8** | **5.4** | **1.1** | **1.5** | **4.4** | **13.5** |
| Gemma3-4B-IT | 8.4 | 9.9 | 17.3 | 17.8 | 3.6 | 5.4 | 4.2 | 6.3 | **2.2** | **2.9** | 17.0 | 24.8 |
| + SFT | **19.6** | **29.8** | **28.6** | **35.7** | **15.3** | **18.8** | **7.5** | **11.2** | 1.2 | 1.7 | **11.4** | **21.0** |
| Gemma3-12B-IT | 36.4 | 40.9 | 36.2 | 39.0 | 11.6 | 14.8 | 12.7 | 19.0 | **10.1** | **13.7** | 29.7 | 37.3 |
| + SFT | **38.9** | **46.3** | **39.3** | **45.5** | **13.1** | **15.8** | **17.9** | **23.1** | 3.7 | 4.6 | **24.4** | **39.0** |

12B) (Team et al., 2025a); and (2) **proprietary models**, including GPT-4o (OpenAI, 2024), Gemini-2.5-Flash (Comanici et al., 2025), and GLM4V-Plus (Team et al., 2025b). Appendix D.2 summarizes the details of each model. All experiments are carried out on a standardized software and hardware stack, using `PyTorch 2.6.0`, `Transformers 4.53.1`, and `CUDA 12.4` on `NVIDIA RTX 5880 Ada Generation 48GB` GPUs. The training and test code is implemented in Python 3.10, with reproducibility ensured by fixed random seeds and deterministic dataloaders. Detailed experimental settings, including prompts, are provided in Appendix D.3 and Appendix E.

## 3.2 SYNTHESIZED TRAINING SET

To synthesize a training set with FlowGen, we systematically define three configurations of parameters representing different levels of difficulty: easy, medium, and hard, as shown in Table 2.

For each renderer (Mermaid, Graphviz, PlantUML, Diagrams), we generate flowcharts across all three difficulty levels. Each difficulty level is combined with 120 topics, with eight random instantiations per topic. This yields a total of 11,520 flowcharts in the synthesized training set.

## 3.3 EXPERIMENTAL RESULTS

### 3.3.1 MAIN RESULTS

Table 3 reports the flowchart parsing results on six benchmarks. *Models fine-tuned on Flow-Gen (+ SFT) generally yield substantial improvements across models and datasets.* For instance, Qwen2.5-VL-3B improves from 12.7% to 51.3% F1 on FlowVQA and from 2.8% to 22.4% on FC_A, while InternVL3-2B increases from 15.7% to 34.6% on FlowVQA. Qwen2.5-VL-7B improves from 43.2% to 60.1% F1 on FlowLearn, demonstrating strong cross-domain generalization rather than overfitting to synthetic data. Importantly, these improvements also manifest under edit similarity. Although performance on the challenging hdBPMN benchmark remains modest, the gains confirm that synthetic training substantially boosts generalization to real-world datasets.

*Moreover, open-source models fine-tuned with FlowGen are comparable to proprietary models on some datasets.* For instance, Qwen2.5-VL-7B attains 60.1% on FlowLearn and 18.3% on hdBPMN, outperforming GLM4V-Plus (31.6% and 8.6%). These findings highlight FlowGen's potential to broaden access to high-performing models with limited resources.

We further evaluate models on flowchart QA benchmarks, as reported in Table 4. All QA evaluations are conducted under zero-shot prompting (see Appendix E). As expected, prompting with gold-standard triplets establishes an upper bound on achievable gains. Notably, triplets extracted by FlowGen consistently achieve performance close to gold-standard annotations across models and datasets, while substantially outperforming triplets extracted by the base model itself. For example, Qwen2.5-VL-3B reaches 88.2% on FlowLearn with FlowGen-extracted triplets, approaching the gold-standard result of 89.1%, and markedly exceeding the 77.1% obtained with self-extracted triplets. Similarly, on AI2D, MiniCPM-V2.6-8B attains

Table 4: Performance comparison (accuracy) of MLLMs on flowchart QA datasets. We compare each model with its variants prompted with gold-standard triplets (+ Gold-Standard Triplets), with triplets extracted by itself (+ Self-Extracted Triplets), and with triplets extracted by Qwen2.5-VL-7B fine-tuned on FlowGen (+ FlowGen-Extracted Triplets).

| Model | FlowVQA | FlowLearn | AI2D | MISS-QA |
|---|---|---|---|---|
| GPT-4o | 90.2 | 83.2 | 71.7 | 63.0 |
| Gemini-2.5-Flash | 88.0 | 83.4 | 63.2 | 67.3 |
| GLM-4V-Plus | 86.4 | 89.8 | 74.6 | 57.0 |
| Qwen2.5-VL-3B | 64.6 | 72.3 | 50.7 | 35.6 |
| + Gold-Standard Triplets | **75.6** | **89.1** | **53.5** | **47.3** |
| + Self-Extracted Triplets | 66.2 | 77.1 | 51.2 | 37.2 |
| + FlowGen-Extracted Triplets | 73.4 | 88.2 | 51.4 | 38.7 |
| Qwen2.5-VL-7B | 74.6 | 71.1 | 60.9 | 42.1 |
| + Gold-Standard Triplets | **83.2** | **73.9** | **62.5** | **55.3** |
| + Self-Extracted Triplets | 77.1 | 70.7 | 60.5 | 44.6 |
| + FlowGen-Extracted Triplets | 81.1 | 68.6 | 61.1 | 49.2 |
| MiniCPM-V2.6-8B | 61.6 | 80.9 | 32.0 | 31.1 |
| + Gold-Standard Triplets | **71.4** | **85.9** | **45.0** | **37.8** |
| + Self-Extracted Triplets | 61.0 | 81.1 | 32.6 | 31.5 |
| + FlowGen-Extracted Triplets | 69.1 | 85.5 | 42.8 | 33.4 |

42.8% using FlowGen-extracted triplets, compared to 45.0% with gold-standard triplets and only 32.6% with self-extraction. Similar trends are observed on MISS-QA and FlowVQA.

Besides triplet-augmented QA, we also fine-tune QA models on FlowGen-synthesized flowchart parsing tasks. Specifically, on FlowVQA and FlowLearn, we fine-tune each model under two settings: (i) SFT on the original QA training set, and (ii) SFT on an equal-sized mixed training set where half of the original QA samples are replaced with FlowGen-synthesized flowchart parsing tasks. As shown in Table 5, incorporating Flow-Gen generally improves end-to-end QA accuracy across models and datasets. For instance, Qwen2.5-VL-7B increases from 85.3% to 89.1% on FlowLearn, while MiniCPM-V2.6-8B improves from 73.2% to 77.6% on FlowVQA. *These gains indicate that FlowGen enhances models'*

Table 5: Performance comparison (accuracy) of MLLMs on flowchart QA datasets. We compare each model with its variants fine-tuned on the original training set (+ SFT on the original training set) and on a mixture of the original training set and FlowGen (+ SFT on a mixed training set).

| Model | FlowVQA | FlowLearn |
|---|---|---|
| Qwen2.5-VL-3B | 64.6 | 72.3 |
| + SFT on the original training set | 81.3 | **85.1** |
| + SFT on a mixed training set | **85.6** | 84.7 |
| Qwen2.5-VL-7B | 74.6 | 71.1 |
| + SFT on the original training set | 88.2 | 85.3 |
| + SFT on a mixed training set | **89.2** | **89.1** |
| MiniCPM-V2.6-8B | 61.6 | 80.9 |
| + SFT on the original training set | 73.2 | 86.2 |
| + SFT on a mixed training set | **77.6** | **87.5** |

*intrinsic ability to understand flowcharts, even when evaluated directly on downstream QA tasks using their fine-tuned checkpoints.* Together with the triplet-augmented QA results, this end-to-end evaluation demonstrates that FlowGen delivers genuine performance improvements by effectively enhancing models' structure-aware visual reasoning capabilities, rather than merely providing auxiliary intermediate representations.

We also investigate whether fine-tuning on FlowGen transfers to broader multimodal reasoning abilities, with detailed results reported in Appendix F.

Table 6:   Ablation results (strict F1) of FlowGen under different settings.

| Training Variant | FlowVQA | CBD | FC_A | FC_B | bdBPMN | FlowLearn |
|---|---|---|---|---|---|---|
| Full FLOWGEN (All Features) | 70.9 | **55.4** | **29.5** | **40.8** | **18.3** | 60.1 |
| w/o Multi-Renderer (Only Mermaid) | **76.2** | 53.3 | 28.5 | 38.1 | 17.0 | **60.9** |
| w/o Nested Subgraphs | 70.0 | 54.6 | 29.2 | 40.6 | 16.6 | 59.2 |
| w/o Split/Merge Arrows | 69.6 | 54.3 | 28.4 | 38.8 | 17.7 | 58.3 |

### 3.3.2   ABLATION STUDY

We perform ablation experiments on Qwen2.5-VL-7B, using synthetic data generated under different constraints while keeping the total training data size identical to the main experiment. We evaluate each variant on six flowchart parsing benchmarks, as reported in Table 6.

(1) **Multi-renderer:** Restricting training data to a single renderer (Mermaid) substantially reduces cross-domain generalization, as shown by drops on CBD (from 55.4% to 53.3%) and FC_B (from 40.8% to 38.1%). In contrast, FlowVQA and FlowLearn slightly improve under this setting, since they exclusively use Mermaid and thus align better with the single-renderer distribution. In general, these results highlight the importance of style variation for learning representations that generalize between renderers and improve robustness.

(2) **Nested subgraphs:** Removing hierarchical nesting leads to markedly reduced accuracy on hdBPMN, the only benchmark among the six that extensively features nested grouping structures, where performance decreases from 18.3% to 16.6%. Smaller declines are also observed on other datasets. This suggests that additional exposure to nested structures provides the model with richer structural signals, enabling it to better capture flowchart organization even when explicit nesting is absent. In general, these results underscore the value of modeling hierarchical layouts.

(3) **Split/merge arrows:** Excluding split/merge arrows results in performance degradation on several benchmarks, including FC_B (from 40.8% to 38.8%), and FlowLearn (from 60.1% to 58.3%). This confirms that such structures are critical for accurately modeling branching logic and parallel workflows, which are common in real-world flowcharts.

To further isolate the contribution of structural information, we perform an additional ablation where all node names and edge labels are replaced with random meaningless strings, preserving the original graph structure while removing semantic annotation. The results are reported in Appendix G.

### 3.3.3   ERROR ANALYSIS

We analyze 50 cases where models fine-tuned with FlowGen fail, including 25 cases from FlowVQA and 25 from hdBPMN. Note that a single failure case may contain multiple error types; the percentages reported in the following can be added to more than 100%. Three main error types emerge.

(1) **Complex nesting (40%):** Deeply nested structures, particularly in hdBPMN, often lead to incorrect handling of hierarchical groupings and subgraph boundaries. Models often flatten or misplace `partOf` relations. Examples are presented in Appendix H.1.

(2) **Edge ambiguity (44%):** For high edge density or overlapping connectors, models sometimes confuse edge directionality or misinterpret relation labels. Examples are presented in Appendix H.2.

(3) **OCR-driven failures (56%):** Although public benchmarks are primarily composed of visually clear diagrams, covering both digitally rendered and handwritten flowcharts, occasional recognition errors still occur (e.g., due to small font or stylized text in node labels). These OCR-related errors lead to incorrect or missing node labels. Illustrative cases are shown in Appendix H.3.

## 4   FLOWGEN FOR MLLMS TESTING

In this section, we construct a challenging test set using FlowGen to evaluate the robustness of MLLMs in flowchart parsing. The evaluation is designed to assess model capabilities in cross-renderer generalization, comprehension of nested structures, and parsing from noisy scanned inputs.

## 4.1 EXPERIMENTAL SETUP

### 4.1.1 DATASETS

The FlowGen test set is designed to evaluate model performance along two challenging dimensions: graph complexity and scanned document degradation. Accordingly, we construct six evaluation subsets by systematically configurating these two factors.

(1) **Graph subsets (Easy/Medium/Hard)**. Following the definition in Section 3.2, the structural difficulty of a flowchart is determined by a set of parameters. These configurations yield three levels of complexity: Easy, Medium, and Hard. For each difficulty level, we synthesize 1,440 flowcharts (4 renderers × 120 topics × 3 instances), resulting in a total of 4,320 flowcharts across the Graph-Easy/Medium/Hard subsets.

(2) **Scanned subsets (Easy/Medium/Hard)**. To evaluate robustness against simulated scanning effects, we further rasterize flowcharts into scanned-like images with degradations such as blur, perspective distortion, and lossy compression using `pillow 10.4.0`[5] and `opencv-python 4.10.0.84`[6]. The severity of these perturbations increases from Easy to Hard. To avoid confounding structural variation, each scanned subset is generated by sampling one instance per topic from Graph-Easy/Medium/Hard. Each subset again contains 1,440 diagrams, leading to 4,320 flowcharts in total. The full configuration ranges for scanned degradations are provided in Appendix I.

The training set is synthesized with the same pipeline and configurations as the test set, ensuring matched distributions, while remaining strictly disjoint to prevent overlap.

### 4.1.2 EVALUATION METRICS

For evaluation on the test set, we follow the strict precision, recall, and F1 score metrics introduced in Section 3.1.2, thus maintaining consistency with the criteria used during the training experiment.

### 4.1.3 PARTICIPATING MODELS

We evaluate the same set of models described in Section 3.1.3.

## 4.2 EXPERIMENTAL RESULTS

### 4.2.1 MAIN RESULTS

Table 7 summarizes model performance on the FlowGen test set. *Overall, we observe that both open-source and proprietary MLLMs struggle on this benchmark*: even state-of-the-art proprietary systems such as GPT-4o and Gemini-2.5-Flash achieve less than 25% F1 on most subsets. Open-source base models also perform poorly, highlighting the structural and stylistic challenges posed by FlowGen. In this experiment, FlowGen-synthesized flowcharts are used for both training and testing as a deliberate stress-test setting. Notably, even under this potentially favorable condition, where renderer-level regularities could in principle be exploited during training, model performance remains far from saturation across all subsets. *This suggests that the challenge of FlowGen primarily arises from its complex and diverse topological structures (e.g., branching, nesting, and heterogeneous layouts), rather than superficial rendering effects.*

We further observe that training on the subset combining samples from all the six subsets (+ SFT on Combined) consistently improves performance across all test subsets, demonstrating a degree of cross-subset generalization. However, this unified training strategy typically underperforms subset-specific fine-tuning, indicating that different subsets capture distinct characteristics. Even with combined supervision, the absolute performance on hard subsets (i.e., Graph-Hard and Scanned-Hard) remains limited. *These results demonstrate that FlowGen constitutes a challenging benchmark for current MLLMs: even when sample distributions overlap and unified training is applied, models still struggle to robustly parse complex flowchart structures, highlighting the diversity and difficulty of the underlying graph distributions.* We also conduct an extra experiment to quantitatively assess the diversity of flowcharts synthesized by FlowGen, with details provided in Appendix J.

---

[5]https://pypi.org/project/pillow/10.4.0/
[6]https://pypi.org/project/opencv-python/4.10.0.84/

Table 7: Performance comparison (strict F1) of MLLMs on FlowGen test subsets.

| Model | Test Subsets | | | | | |
|---|---|---|---|---|---|---|
| | Graph-Easy | Graph-Medium | Graph-Hard | Scanned-Easy | Scanned-Medium | Scanned-Hard |
| GPT-4o | 38.4 | 16.2 | 12.0 | 24.0 | 23.2 | 20.5 |
| Gemini-2.5-Flash | 43.2 | 13.7 | 9.7 | 22.9 | 23.1 | 21.6 |
| GLM-4V-Plus | 30.0 | 9.4 | 5.3 | 15.3 | 19.3 | 18.3 |
| Qwen2.5-VL-3B | 20.4 | 6.2 | 2.9 | 10.2 | 10.0 | 9.0 |
| + SFT on Graph Easy | **50.2** | 24.0 | 16.0 | 33.0 | 30.5 | 29.0 |
| + SFT on Graph Medium | 36.3 | **32.9** | 28.2 | 33.1 | 33.1 | 31.3 |
| + SFT on Graph Hard | 26.9 | 30.5 | **30.3** | 29.4 | 29.8 | 28.4 |
| + SFT on Scanned Easy | 45.1 | 31.8 | 29.0 | 34.8 | 35.3 | 34.1 |
| + SFT on Scanned Medium | 43.6 | 32.2 | 28.5 | 34.8 | 35.5 | 33.8 |
| + SFT on Scanned Hard | 43.7 | 32.6 | 29.3 | 35.6 | 35.5 | **35.0** |
| + SFT on Combined | 49.9 | 31.0 | 24.6 | **36.3** | 35.7 | 33.1 |
| Qwen2.5-VL-7B | 35.7 | 10.9 | 7.7 | 18.9 | 18.5 | 17.2 |
| + SFT on Graph Easy | **74.9** | 35.0 | 21.2 | 45.3 | 44.3 | 41.9 |
| + SFT on Graph Medium | 67.3 | **52.4** | **44.0** | 56.2 | 55.3 | 52.3 |
| + SFT on Graph Hard | 39.1 | 43.6 | 43.3 | 42.6 | 42.9 | 40.8 |
| + SFT on Scanned Easy | 66.8 | 47.8 | 41.6 | 53.9 | 53.1 | 49.7 |
| + SFT on Scanned Medium | 68.8 | 48.1 | 42.0 | 54.4 | 53.6 | 51.0 |
| + SFT on Scanned Hard | 66.0 | 47.1 | 42.0 | 53.0 | 52.6 | 50.2 |
| + SFT on Combined | 66.5 | 47.2 | 42.6 | 52.8 | 51.7 | 50.2 |
| MiniCPM-V2.6-8B | 7.3 | 2.1 | 2.1 | 4.0 | 4.3 | 3.5 |
| + SFT on Graph Easy | 33.0 | 13.1 | 9.0 | 18.8 | 18.1 | 17.7 |
| + SFT on Graph Medium | 23.9 | 19.6 | 15.2 | 20.0 | 19.9 | 18.1 |
| + SFT on Graph Hard | 17.5 | 17.4 | 17.4 | 17.8 | 17.9 | 16.8 |
| + SFT on Scanned Easy | 29.6 | 19.1 | 16.5 | 22.2 | 22.1 | 20.8 |
| + SFT on Scanned Medium | 29.6 | 19.0 | 16.1 | 22.3 | 22.3 | 20.4 |
| + SFT on Scanned Hard | 27.1 | 18.0 | 16.3 | 21.3 | 20.4 | 19.4 |
| + SFT on Combined | **39.5** | **24.2** | **20.6** | **29.0** | **28.8** | **27.1** |

### 4.2.2 ERROR ANALYSIS

We analyze 50 failure cases on the FlowGen's test set and identify four major error categories: Cross-renderer generalization, Complex nesting, Edge ambiguity, and OCR-driven failures. Compared to previous datasets, FlowGen highlights an additional challenge—**Cross-renderer generalization**, where stylistic differences between renderers (e.g., PlantUML vs. Mermaid) frequently lead to structural misinterpretations. Detailed error breakdowns and examples are provided in Appendix K.

## 5 CONCLUSION

In this work, we present FlowGen, a controllable flowchart synthesizer that exposes explicit structural configurations and supports diverse rendering styles, providing both a scalable training resource and a systematic test benchmark. Our experiments show that current MLLMs, including leading proprietary models, perform poorly on the FlowGen test set, particularly under high structural complexity and varied rendering styles. However, fine-tuning on FlowGen substantially boosts the robustness and generalization of open-source models, highlighting controllable synthesis as an effective strategy for advancing flowchart understanding and its downstream reasoning tasks. By releasing FlowGen, we provide the community with a reproducible and extensible framework for training, testing, and advancing models for graph-structured visual reasoning.

To address the limitations of our work, our future work includes exploring diverse implementations of split/merge arrows beyond introducing virtual nodes, incorporating more features of real flowcharts into our synthesizer pipeline, and experimenting with more complex flowchart-based tasks such as root cause analysis in IT operations.

REPRODUCIBILITY STATEMENT

The detailed experimental settings of synthesizer, models, hyper-parameter settings, and computational resources can be found in Section 3.1.3, Appendix D and Appendix E. The codes and datasets for reproducing our evaluation results are provided in the GitHub repository.

ACKNOWLEDGEMENTS

This work was supported by the National Science and Technology Innovation 2030 Major Program (2025ZD0544900) and sponsored by CCF-Lenovo Blue Ocean Research Fund.

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

## A  THE USE OF LARGE LANGUAGE MODELS (LLMs)

As indicated in the submission form, we answer "Large Language Models: Yes, to aid or polish writing." LLMs were only used for language refinement, such as polishing wording, improving clarity, and ensuring consistency. They were not involved in research ideation, methodology design, or experimental analysis. All scientific contributions and results are solely due to the authors.

## B  RELATED WORK

### B.1  EXISTING DATASETS

Research on flowchart understanding has traditionally relied on fixed datasets rather than controllable synthesizers. Existing datasets such as IconQA (Lu et al., 2022), FlowchartQA (Tannert et al., 2023), FlowVQA (Singh et al., 2024), and FlowLearn (Pan et al., 2024) provide important benchmarks for evaluating models on tasks ranging from flowchart parsing to flowchart QA. Other datasets such as CBD (Bhushan & Lee, 2022), FC_A (Awal et al., 2011), FC_B (Bresler et al., 2016), and hdBPMN (Schäfer et al., 2021b) provide domain-specific diagram test resources. However, these resources share several limitations. First, they lack controllability: users cannot systematically vary structural properties such as node count, edge density, or nesting depth. Second, stylistic diversity is limited, since most datasets rely on a single renderer or annotation style, whereas real-world flowcharts come from heterogeneous sources. Finally, most existing datasets are designed for evaluation only, and do not support large-scale, controllable synthesis for training purposes. To date, there are no general-purpose flowchart synthesizers that can flexibly generate both training and test sets with controllable complexity and rendering diversity, leaving an important gap in the field. Table 1 provides a systematic comparison between our proposed synthesizer and existing datasets. Our FlowGen is unique in combining controllable structural features, multi-renderer diversity, and scalability to both training and test, filling a critical gap in the field.

### B.2  EXISTING METHODS

Alongside datasets, a range of methods have been explored for flowchart-based visual reasoning. Early approaches adopt a detection-and-association paradigm, such as Arrow R-CNN (Schäfer et al., 2021a) and FR-DETR (Sun et al., 2022), which first localize symbols and arrows before reconstructing the underlying graph. While these pipelines can handle relatively simple diagrams, they typically depend on handcrafted heuristics and degrade under high structural complexity or stylistic variation. More recent work has shifted toward end-to-end VLMs and MLLMs, including BLIP-2 (Li et al., 2023), LLaVA (Liu et al., 2023), PaLI-X (Chen et al., 2023), and Qwen-VL (Bai et al., 2023). These models integrate perception and reasoning within a unified framework, achieving strong performance on natural-image VQA tasks. However, they remain limited in structural parsing of flowcharts, where precise recovery of hierarchical and relational information is required. In this work, we focus exclusively on evaluating representative MLLMs. By systematically analyzing their performance on controlled test datasets, we aim to reveal their strengths, limitations, and the specific challenges posed by flowchart understanding.

## C  APPLICATION DOMAINS AND SEMANTIC VOCABULARIES

This appendix provides additional details on semantic annotation in the construction of FlowGen. Specifically, FlowGen covers 120 distinct application domains, spanning a wide range of real-world scenarios. For each domain, we curate a vocabulary consisting of 40 node names and 40 edge labels, which are initially generated using GPT-4o. These domain-specific vocabularies are used to annotate graph nodes and edges, ensuring semantic cohesiveness within graphs and domain diversity across graphs. Representative examples from several domains are presented below. The complete lists are available in /Synthesizer/my_dictionary.py in our GitHub repository.

```
Application Domains and Semantic Vocabularies

APPLICATIONS = [
    'Finance', 'Healthcare', 'Manufacturing',
    'Education', 'E_commerce', 'Transportation', 'Energy',
    'Legal', 'Logistics',  'Telecom', 'Agriculture','Tourism',
    'Real Estate','Public Safety','Entertainment',
    'Environment','Government', 'Retail', 'Automotive',
    'Aerospace',  'Insurance',  'Cybersecurity', 'Smart Home',
    'Smart City', 'Biotechnology', 'Pharmaceuticals', 'Robotics',
    'Construction', 'Media', 'Sports', 'Fashion', 'Human Resources',
    ...
]

NODE_NAMES = {
    'Finance': [
        'Account Verification', 'Credit Scoring',
        'Approval Decision', 'Fraud Check', 'Transaction Audit',
        'Investment Planning', 'Financial Advisory',
        ...
    ],
    'Healthcare': [
        'Patient Intake', 'Lab Test', 'Diagnosis', Prescription',
        'Medical History', 'Health Screening', 'Allergy Test',
        'Genetic Counseling', 'Pain Management', 'Patient Transfer',
        ...
    ],
    ...
}

EDGE_LABELS = {
    'Finance': [
        'verifies', 'approves', 'rejects', 'checks',
        'calculates', 'recommends', 'consults', 'plans',
        'allocates', 'monitors', 'models', 'settles',
        ...
    ],
    'Healthcare': [
        'diagnoses', 'examines', 'tests', 'treats', 'scans',
        'monitors', 'records', 'administers', 'researches',
        'prevents', 'educates', 'surveys', 'immunizes',
        ...
    ],
    ...
}
```

## D  IMPLEMENTATION DETAILS

### D.1  SUMMARY OF EVALUATION DATASETS

Table 8 summarizes the public datasets used in our evaluation experiments, providing their task, size, source, and license to facilitate reproducibility and transparency.

### D.2  MODEL VERSIONS AND SOURCES

Table 9 summarizes the proprietary and open-source MLLMs evaluated in this study, organized by provider with release dates and official access sources. Open-source models are linked to their Hugging Face repositories, while proprietary models are accessed through their respective APIs.

Table 8: Summary of evaluation datasets used in our experiments.

| Dataset | Task | Size | Source | License |
|---|---|---|---|---|
| FlowVQA | Parsing & QA | 956 | Synthetic flowcharts | MIT License |
| CBD | Parsing | 96 | Computerized block diagrams | Research-only |
| FC_A | Parsing | 145 | Hand-written flowcharts | CC-BY-NC-SA |
| FC_B | Parsing | 196 | Hand-written flowcharts | NA |
| hdBPMN | Parsing | 173 | Hand-drawn BPMN diagrams | CC-BY-4.0 |
| FlowLearn | Parsing & QA | 2,000 | Scientific literature & Synthetic | CC-BY-NC-SA-4.0 |
| AI2D | QA | 4,060 | Scientific Diagrams | CC BY-SA |
| MISS-QA | QA | 173 | Schematic Diagrams | CC-BY-4.0 |

Table 9: Evaluated proprietary and open-source MLLMs.

| Provider | Model | Release | Source |
|---|---|---|---|
| *Proprietary Multimodal Large Language Models* | | | |
| OpenAI | GPT-4o | 2024-08 | OpenAI API |
| Google | Gemini-2.5-Flash | 2025-04 | Google AI Studio API |
| Zhipu AI | GLM-4V-Plus | 2024-08 | Zhipu AI API |
| *Open-Source Multimodal Large Language Models* | | | |
| Alibaba | Qwen2.5-VL-3B | 2025-01 | https://huggingface.co/Qwen/Qwen2.5-VL-3B-Instruct |
| | Qwen2.5-VL-7B | 2025-01 | https://huggingface.co/Qwen/Qwen2.5-VL-7B-Instruct |
| OpenGVLab | InternVL3-2B | 2025-01 | https://huggingface.co/OpenGVLab/InternVL3-2B |
| OpenBMB | MiniCPM-V2.6-8B | 2025-04 | https://huggingface.co/openbmb/MiniCPM-V-2_6 |
| LLaVA Team | LLaVA-v1.6-Mistral-7B | 2024-12 | https://huggingface.co/llava-hf/llava-v1.6-mistral-7b-hf |
| Google | Gemma3-4B | 2025-03 | https://huggingface.co/google/gemma-3-4b-it |
| | Gemma3-12B | 2025-03 | https://huggingface.co/google/gemma-3-12b-it |

## D.3 FINE-TUNING HYPERPARAMETERS

For open-source models, we perform supervised fine-tuning (SFT) on FlowGen-synthesized training set using the configuration shown in Table 10. The training employs LoRA adapters on both language and vision components, with other modules frozen for efficiency.

Table 10: Hyperparameter configuration for SFT on FlowGen-synthesized training set.

| Hyperparameter | Value |
|---|---|
| Framework | PyTorch 2.6.0 + DeepSpeed (ZeRO-3) |
| Precision | BF16 (FP16 disabled) |
| Gradient checkpointing | Enabled |
| Optimizer | AdamW ($\beta_1 = 0.9, \beta_2 = 0.999$, weight decay=0.1) |
| Learning rate | $5 \times 10^{-4}$ (cosine scheduler) |
| Warmup ratio | 0.03 |
| Global batch size | 32 |
| Per-device batch size | 4 |
| Gradient accumulation steps | 8 |
| Epochs | 1 |
| LoRA rank / alpha / dropout | 64 / 64 / 0.05 |
| Vision tower | Frozen (with vision LoRA enabled) |
| LLM backbone | Frozen |
| Merger | Frozen |
| Image resolution range | $256 \times 28 \times 28 - 1280 \times 28 \times 28$ |

This configuration ensures efficient training with limited GPU resources while still adapting both textual and visual representations through LoRA.

# E  PROMPT TEMPLATES

---

### Prompt Template for Flowchart Parsing

**[System Input]**

You are an expert assistant specialized in flowchart understanding.
Your task is to parse flowchart images into structured knowledge triples that capture both topology and semantics.

**[User Input]**

{image input: flowchart}

{text input:  <image>Please describe all the information in the flowchart image in the form of triples:
For each arrow labeled with X directed from node A to node B, output a triple: <A, X, B>;
For each unlabeled arrow directed from node A to node B, output a triple: <A, connectedTo, B>;
For each node A fully inside node group B, output a triple <A, partOf, B>.

The following triples are example outputs:
<Food Packaging Improvement, secures, Quantum Computing Integration>
<Graphene Production, catalyzes, Nanosensors>
<Graphene Production, partOf, Drug Delivery Systems>
<Nanocomposites, connectedTo, Spectroscopy>
<Nanodevice Fabrication, connectedTo, Graphene Production>
<Nanosensors, connectedTo, Nanocomposites>
<Spectroscopy, educates, Nanodevice Fabrication>}

---

### An Example Prompt for Answering Flowchart-based Yes/No Questions

**[System Input]**

You are an expert assistant specialized in flowchart understanding.
Your task is to parse flowchart images into structured knowledge triples that capture both topology and semantics.

**[User Input]**

{image input: flowchart}

{text input:  Determine whether the following description of the image is correct (just answer Yes or No) :

"Consistent failures through this type of intervention constitute a challenging group for the perception model as seen on the right." connectedto "Perception Model is used for discovery of failures at the scene level."

The following description, extracted in the form of triples from the image, is provided for reference and may contain errors.
<BERT, connectedTo, Coca-Cola Truck>
<Coca-Cola Truck, connectedTo, Perception Model>
<Coca-Cola Truck, connectedTo, Perception Model>
......
<Police Car, partOf, Interventions>
<Police Car, partOf, Scene Understanding>
<Scene Understanding, connectedTo, Police Car>"

The answer is: }

## An Example Prompt for Answering Flowchart-based Multiple-Choice Questions

**[System Input]**

You are an expert assistant specialized in flowchart understanding.
Your task is to parse flowchart images into structured knowledge triples that capture both topology and semantics.

**[User Input]**

{image input: flowchart}

{text input: <image>Please answer the following multiple-choice question about the image:

The sexual or asexual process by which organisms generate new individuals of the same kind is called?
(A).reproduction
(B).pollination
(C).growth
(D).germination

The following description, extracted in the form of triples from the image, is provided for reference and may contain errors.
<Fertilization, connectedTo, Green Bean>
<Growth, connectedTo, Germination>
<Growth, connectedTo, Reproduction>
......
<Reproduction, connectedTo, Pollination>
<Reproduction, connectedTo, Growth>
<Green Bean, connectedTo, Germination>

The answer is: }

## Prompt for Topic and Node/Edge Names Generation

**[System Input]**

You are tasked with helping build a synthetic flowchart dataset. Please generate 120 distinct application domains (topics).
For each topic:
(1) Provide the topic name.
(2) List 40 concise node names that could appear in flowcharts within this topic.
(3) List 40 concise edge names that describe interactions or relationships among the nodes.

Constraints:
- Use short phrases (1–4 words) for node and edge names.
- Avoid redundancy across different topics.
- Ensure that node names are realistic to the chosen topic.
- Ensure that edge names reflect meaningful relationships.
- Output in structured JSON format:

**[GPT-4o Output]**

```
{
{ "Topic": "Finance",
"Nodes names": ["Account Verification", "Credit Scoring", "Risk Assessment" ...],
"edges names": ["verifies", "approves", "budgets", ...] },
{ "Topic": "Healthcare",
"Node names": ["Patient Intake", "Medical History", "Emergency Response", ...],
"Edge names": ["diagnoses", "treats", "examines", ...] },
......
}
```

## F   EXPERIMENTS ON MMMU

Recent studies have shown that the composition of SFT data plays a critical role in determining the resulting capabilities of multimodal models. In particular, Dong et al. (2024) demonstrate that naively mixing heterogeneous tasks often leads to optimization conflicts, while sequential fine-tuning can induce catastrophic forgetting. Motivated by these findings, and following their interleaved training paradigm, we investigate whether fine-tuning on FlowGen transfers beyond flowchart parsing to broader multimodal reasoning.

Specifically, we examine whether incorporating FlowGen-synthesized flowchart parsing tasks during SFT improves performance on MMMU (Yue et al., 2024), a diverse benchmark covering general-domain multimodal reasoning tasks that are not directly related to flowcharts. This experiment aims to assess whether FlowGen only induces task-specific gains, or it promotes more generalizable structural reasoning abilities.

Since MMMU does not provide an official training split, we follow common practice and use its 900 validation samples for SFT and 150 development samples as test cases. For each model, we consider two SFT settings:

- Fine-tuning solely on MMMU samples.
- Fine-tuning on an equal-sized mixed dataset, where 50% of MMMU samples are replaced with FlowGen-synthesized flowchart parsing samples. We adopt an interleaved training strategy at the batch level, ensuring that FlowGen and MMMU samples are jointly optimized throughout training.

Table 11: Performance comparison (accuracy) of MLLMs on MMMU. We compare each model with its variants fine-tuned on the original training set (+ SFT on the original training set) and on a mixture of the original training set and FlowGen (+ SFT on a mixed training set).

| Training Variant | Qwen2.5-VL-3B | Qwen2.5-VL-7B | MiniCPM-V2.6-8B |
|---|---|---|---|
| Base Model | 53.3 | 58.6 | 49.4 |
| + SFT on the original training set | 59.3 | 62.6 | 53.3 |
| + SFT on a mixed training set | **60.2** | **65.3** | **54.0** |

As shown in Table 11, models fine-tuned with the mixed dataset consistently outperform their MMMU-only counterparts on the MMMU benchmark. Notably, these improvements are observed despite replacing half of the MMMU supervision with FlowGen data, indicating that FlowGen does not dilute task-relevant learning but instead provides complementary signals.

We attribute this behavior to the structural nature of FlowGen supervision. FlowGen emphasizes explicit modeling of entities, relations, and spatial layouts through triplet extraction, which encourages the model to learn transferable representations of graph structure and visual organization. Such inductive biases are broadly applicable across multimodal reasoning tasks, including those in MMMU that require compositional understanding, spatial grounding, or multi-step inference.

# G    ABLATION ON SEMANTIC ANNOTATION

To further isolate the contribution of structural information in flowchart parsing, we conduct an additional ablation in which all node names and edge labels are replaced with random meaningless strings. Specifically, node names are substituted with random strings starting with uppercase letters, while edge labels are replaced with random lowercase strings of length 5–12. This preserves minimal formatting conventions while eliminating meaningful semantics. This experiment evaluates whether FlowGen-trained models can still leverage structural cues when textual semantics are removed.

Table 12: Ablation results (strict F1) of FlowGen using semantic annotation (+ SFT on Full Flow-Gen) or random strings (+ SFT on FlowGen w/o Semantic Annotation) for node names and edge labels.

| Model | FlowVQA | CBD | FC_A | FC_B | hdBPMN | FlowLearn |
|---|---|---|---|---|---|---|
| Qwen2.5-VL-3B | 12.7 | 17.9 | 2.8 | 8.6 | 2.3 | 20.6 |
| + SFT on Full FlowGen | **51.3** | **49.8** | **22.4** | **34.6** | **8.4** | **41.6** |
| + SFT on FlowGen w/o Semantic Annotation | 50.9 | 44.1 | 13.6 | 24.1 | 5.9 | 39.5 |
| Qwen2.5-VL-7B | 59.4 | 49.1 | 25.8 | 32.0 | 16.6 | 43.2 |
| + SFT on Full FlowGen | **70.9** | **55.4** | **29.5** | **40.8** | **18.3** | **60.1** |
| + SFT on FlowGen w/o Semantic Annotation | 64.5 | 52.9 | 28.4 | 34.8 | 17.4 | 56.0 |
| MiniCPM-V2.6-8B | 17.6 | 20.4 | 7.5 | 8.8 | 3.4 | 9.1 |
| + SFT on Full FlowGen | **44.0** | **37.9** | **13.4** | **22.9** | **3.6** | **20.0** |
| + SFT on FlowGen w/o Semantic Annotation | 35.2 | 29.8 | 9.7 | 13.9 | 3.6 | 19.1 |

Table 12 reports the results. Compared to models trained with FlowGen's original random but semantically meaningful labels, performance with meaningless labels shows a moderate degradation. Importantly, the performance gains of SFT remain consistent and significant across different models and datasets. *These results indicate that the benefits from using FlowGen are primarily driven by learning structural patterns, topological relations, and spatial layouts from flowcharts, rather than relying solely on textual semantics.*

# H ERROR ANALYSIS ON PUBLIC DATASETS

## H.1 COMPLEX NESTING ERROR ON PUBLIC DATASET

---

### An Example of Complex Nesting Error on hdBMPN Dataset

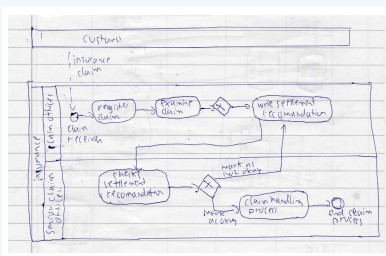

**[System Input]**

You are an expert assistant specialized in flowchart understanding.
Your task is to parse flowchart images into structured knowledge triples that capture both topology and semantics.

**[User Input]**

{image input: flowchart}

{text input: <image>Please describe all the information in the flowchart image in the form of triples:
For each arrow labeled with X directed from node A to node B, output a triple: <A, X, B>;
For each unlabeled arrow directed from node A to node B, output a triple: <A, connectedTo, B>;
For each node A fully inside node group B, output a triple <A, partOf, B>.

The following triples are example outputs:
<Food Packaging Improvement, secures, Quantum Computing Integration>
<Graphene Production, catalyzes, Nanosensors>
<Graphene Production, partOf, Drug Delivery Systems>
<Nanocomposites, connectedTo, Spectroscopy>
<Nanodevice Fabrication, connectedTo, Graphene Production>
<Nanosensors, connectedTo, Nanocomposites>
<Spectroscopy, educates, Nanodevice Fabrication>}

---

**Qwen2.5-VL-7B Output:**
<Claim Handling Process, connectedTo, End Claim Payers>
<Claim Handling Process, mark as Not Okay, Write Settlement Recommendation>
<Check Settlement Recommendation, connectedTo, Claim Handling Process>
<Customer Insurance Claim, connectedTo, Senior Claim Officer>
<Customer Insurance Claim, connectedTo, Write Settlement Recommendation>
<Examine Claim, connectedTo, Check Settlement Recommendation>
<Examine Claim, connectedTo, Write Settlement Recommendation>
<End Claim Payers, connectedTo, Claim Handling Process>
<Rewrites Claim, connectedTo, Examine Claim>
<Senior Claim Officer, claim receives, Rewrites Claim>
<Write Settlement Recommendation, connectedTo, Check Settlement Recommendation>

---

**Error Analysis:**
Qwen2.5-VL-7B's prediction fails to capture the hierarchical containment between roles and tasks. In the gold standard, operations such as "register claim", "examine claim", and "write settlement recommendation" are explicitly nested under "claim officer" and further within "insurance". The model instead flattens this structure, inventing entities like "End Claim Payers" and producing spurious links (e.g., ⟨Senior Claim Officer, claim receives, Rewrites Claim⟩), while omitting key *partOf* relations. This collapse of nested dependencies leaves the explanation structurally inconsistent and semantically incomplete.

## H.2 EDGE AMBIGUITY ERROR ON PUBLIC DATASET

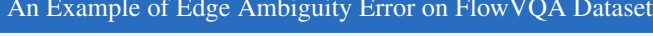

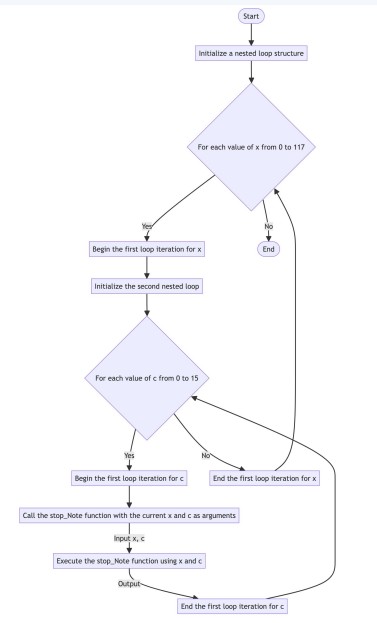

**[System Input]**

Configured as in Appendix H.1

**[User Input]**

Configured as in Appendix H.1

**Qwen2.5-VL-7B Output:**
<Begin the first loop iteration for c, connectedTo, Call the stop_Note function with the current x and c as arguments>
<Begin the first loop iteration for x, connectedTo, Initialize the second nested loop>
<Call the stop_Note function with the current x and c as arguments, Input x, c, Execute the stop_Note function using x and c>
<Execute the stop_Note function using x and c, Output, End the first loop iteration for c>
<End, No, For each value of x from 0 to 117>
<End the first loop iteration for c, connectedTo, For each value of c from 0 to 15>
<End the first loop iteration for x, connectedTo, For each value of c from 0 to 15>
<For each value of c from 0 to 15, Yes, Begin the first loop iteration for c>
<For each value of x from 0 to 117, Yes, Begin the first loop iteration for x>
<Initialize a nested loop structure, connectedTo, For each value of x from 0 to 117>
<Initialize the second nested loop, connectedTo, For each value of c from 0 to 15>
<Start, connectedTo, Initialize a nested loop structure>

**Error Analysis:**
Qwen2.5-VL-7B's prediction suffers from edge ambiguity, misrepresenting the direction and semantics of key transitions in the flow loop. In the gold standard, control flow edges clearly encode how each loop iteration branches back to its condition (e.g., ⟨End the first loop iteration for c, connectedTo, For each value of c from 0 to 15⟩) and how termination is handled (e.g., ⟨For each value of x from 0 to 117, No, End⟩). The model instead reverses or distorts these relations, producing edges such as ⟨End, No, For each value of x from 0 to 117⟩ and ⟨End the first loop iteration for x, connectedTo, For each value of c from 0 to 15⟩, which invert the intended logic. While many node labels are preserved, the predicted edges fail to capture whether they represent continuation, termination, or loop-back connections, resulting in a structurally inconsistent flow and an inaccurate representation of the looping process.

## H.3 OCR Driven Error on Public Dataset

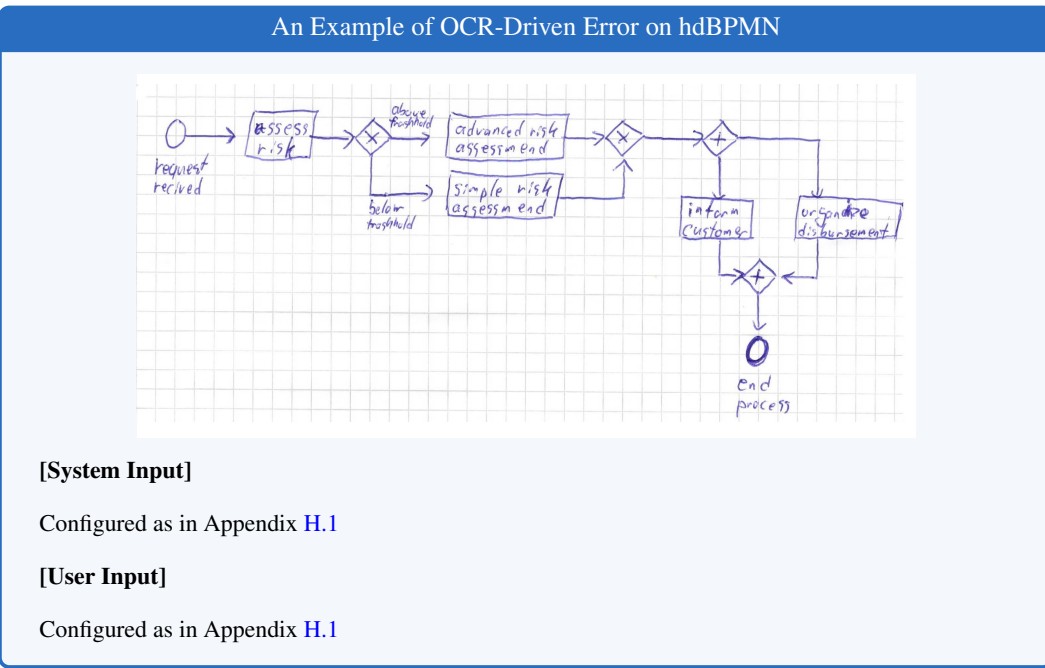

**An Example of OCR-Driven Error on hdBPMN**

**[System Input]**

Configured as in Appendix H.1

**[User Input]**

Configured as in Appendix H.1

**Qwen2.5-VL-7B Output:**
<Advanced Risk Assessment, connectedTo, Simple Risk Assessment>
<Assess Risk, above threshold, Advanced Risk Assessment>
<Assess Risk, below threshold, Simple Risk Assessment>
<Advanced risk asessmend, connectedTo, Intram customer>
<Customer Notification, connectedTo, Urgent Wire Disbursement>
<Simple Risk Assessment, connectedTo, Customer Notification>
<Urgent Wire Disbursement, connectedTo, End Procesy>
<End Procesy, connectedTo, Customer Notification>
<Request Received, connectedTo, Assess Risk>

**Error Analysis:**
Qwen2.5-VL-7B's outputs exhibit typical OCR-driven errors: misrecognized text such as "intram customer" (likely intended as "inform customer") and "End Procesy" (likely "End Process") results in incorrect node labels like "Customer Notification" and "Urgent Wire Disbursement" that deviate from the source diagram. Consequently, the model generates spurious edges (e.g., ⟨End Procesy, connectedTo, Customer Notification⟩) while omitting valid dependencies, such as the direct sequence from "Inform customer" to "End Process" implied by the original workflow. These OCR-induced distortions compromise both the semantics and the topology of the predicted graph, illustrating how recognition noise propagates directly into structural parsing errors.

## I Scanned Style Configurations

Table 13 summarizes the configuration used to synthesize scanned-style flowcharts with three difficulty levels.

Table 13: Configurations for scanned-style flowcharts generation with three difficulty levels.

| Difficulty | Blur radius | Vignette | Rotation | Perspective | Color tint | Noise level | JPEG quality |
|---|---|---|---|---|---|---|---|
| Easy | 0.8 | 0.1 | 1.0° | No | Yellowish | 0.005 | 90 |
| Medium | 1.0 | 0.3 | 1.2° | No | Yellowish | 0.008 | 70 |
| Hard | 1.2 | 0.5 | 1.5° | Yes | Gray | 0.015 | 40 |

## J    ANALYSIS OF INTRA-TOPIC FLOWCHART DIVERSITY

To quantitatively assess the diversity of graph structures synthesized under the same topic, we measure the structural dissimilarity between flowcharts using graph edit distance (GED). GED is defined as the minimum number of elementary graph operations, including node or edge insertion, deletion, and substitution, required to transform one graph into another.

Table 14: Mean graph edit distance (GED) with standard deviation (SD) over intra-topic graphs constructed by FlowGen.

|  | Mean GED | SD |
| --- | --- | --- |
| ***Within the same difficulty level*** |  |  |
| Easy | 18.0 | 2.2 |
| Medium | 28.5 | 4.1 |
| Hard | 36.8 | 6.4 |
| ***Between difficulty levels*** |  |  |
| Easy-Medium | 38.0 | 5.1 |
| Easy-Hard | 76.8 | 8.9 |
| Medium-Hard | 48.5 | 7.2 |

Table 14 reports the mean GED with standard deviation (SD) between flowcharts from the same topic, evaluated both within the same difficulty level and between different difficulty levels. The results show substantial structural variation among intra-topic flowcharts: mean GED values range from 18.0 to 36.8 for flowcharts of the same difficulty level, and further increase to 76.8 when comparing flowcharts across difficulty levels. *These findings demonstrate that FlowGen produces highly diverse flowchart structures even within a single topic, supporting the diversity and challenge of the test subsets.*

## K    ERROR ANALYSIS ON FLOWGEN TEST SET

We further analyze 50 failure cases on FlowGen's test sets, which incorporates renderer diversity and simulated scanning artifacts. Note that a single failure case may exhibit multiple error types, so the percentages reported below can sum to more than 100%. Four major error types emerge.

(1) **Cross-renderer generalization (48%):** Visual conventions differ substantially across renderers (e.g., PlantUML vs. Mermaid). As a result, models often fail to generalize across styles, misinterpreting arrowheads, connector boundaries, or nesting layouts. Such renderer-specific artifacts cause structural discrepancies even when the underlying abstract topology is identical. Representative cases are shown in Appendix K.1.

(2) **Complex nesting (38%):** Similar to the errors observed on other benchmarks, models frequently mishandle deeply nested `partOf` relations. Examples are provided in Appendix K.2.

(3) **Edge ambiguity (54%):** When multiple edges overlap or connectors are visually congested, models still confuse edge directionality or labels. These cases resemble the ambiguity observed in Section 3.3.3 but are further exacerbated by renderer-induced variations. Examples are given in Appendix K.3.

(4) **OCR-driven failures (66%):** In scanned test subsets, text recognition errors often corrupt node labels or generate extra nodes that do not exist in the original flowchart. Examples are provided in Appendix K.4.

### K.1 CROSS-RENDERER ERROR ON FLOWGEN TEST SET

---

**An Example of Cross-Renderer Error on FlowGen Test Set**

[System Input]

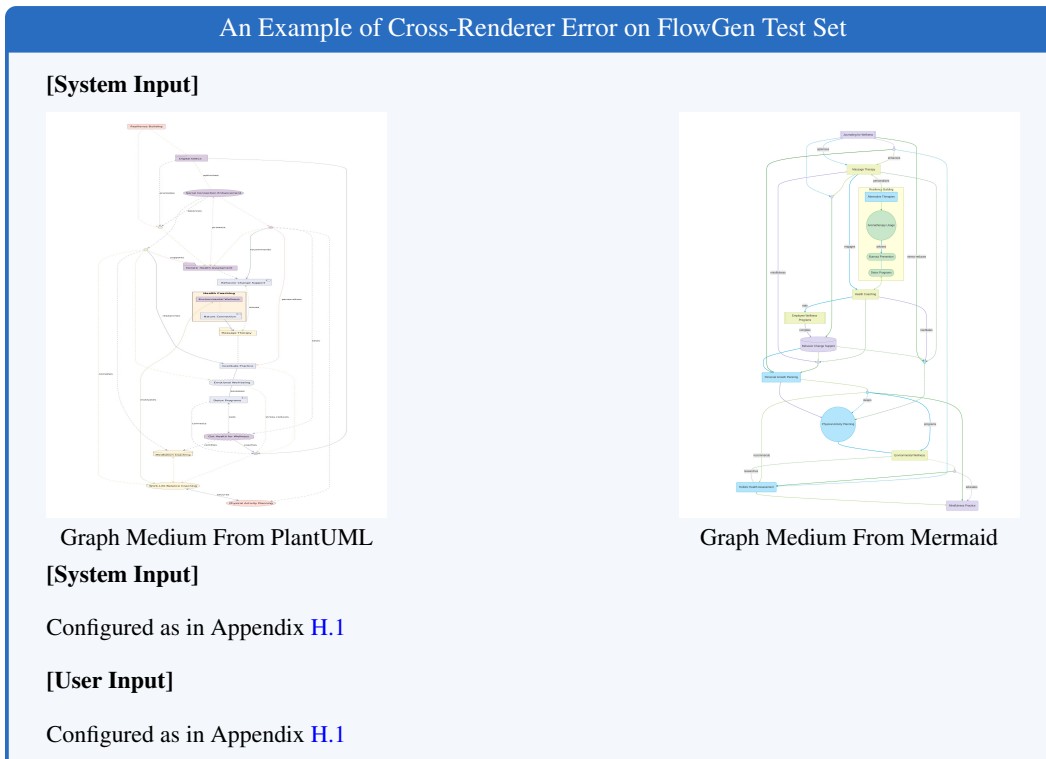

Graph Medium From PlantUML        Graph Medium From Mermaid

[System Input]

Configured as in Appendix H.1

[User Input]

Configured as in Appendix H.1

---

**Qwen2.5-VL-7B For PlantUML Flowchart Output:**
<Behavior Change Support, connectedTo, Holistic Health Assessment><Behavior Change Support, protects, Social Connection Enhancement><Detox Programs, assesses, Emotional Well-being><Detox Programs, eats, Gut Health for Wellness><Detox Programs, moves, Environmental Wellness><Detox Programs, stress-reduces, Physical Activity Planning><Digital Detox, connectedTo, Resilience Building><Digital Detox, optimizes, Social Connection Enhancement><Digital Detox, promotes, Holistic Health Assessment><Digital Detox, tests, Physical Activity Planning><Emotional Well-being, connectedTo, Gratitude Practice><Environmental Wellness, connectedTo, Nature Connection><Environmental Wellness, partOf, Health Coaching><Gut Health for Wellness, certifies, Work-Life Balance Coaching><Gut Health for Wellness, coaches, Physical Activity Planning><Holistic Health Assessment, connectedTo, Behavior Change Support><Holistic Health Assessment, connectedTo, Social Connection Enhancement><Meditation Coaching, connectedTo, Work-Life Balance Coaching><Meditation Coaching, motivates, Holistic Health Assessment><Massage Therapy, connectedTo, Gratitude Practice><Massage Therapy, partOf, Health Coaching><Nature Connection, connectedTo, Environmental Wellness><Nature Connection, partOf, Health Coaching><Physical Activity Planning, connectedTo, Digital Detox><Physical Activity Planning, connectedTo, Work-Life Balance Coaching><Physical Activity Planning, eats, Detox Programs><Resilience Building, complies, Quantum Computing Integration><Resilience Building, connectedTo, Digital Detox><Resilience Building, connectedTo, Holistic Health Assessment><Resilience Building, connectedTo, Meditation Coaching><Resilience Building, connectedTo, Physical Activity Planning><Resilience Building, personalizes, Physical Activity Planning><Resilience Building, researches, Gratitude Practice><Resilience Building, supports, Holistic Health Assessment><Social Connection Enhancement, balances, Holistic Health Assessment><Social Connection Enhancement, connectedTo, Holistic Health Assessment><Social Connection Enhancement, optimizes, Digital Detox><Social Connection Enhancement, protects, Behavior Change Support><Social Connection Enhancement, recommends, Behavior Change Support><Work-Life Balance Coaching, connectedTo, Meditation Coaching><Work-Life Balance Coaching, connectedTo, Physical Activity Planning><Work-Life Balance Coaching, connects, Detox Programs><Work-Life Balance Coaching, secures, Physical Activity Planning>

---

**Qwen2.5-VL-7B For Mermaid Flowchart Output:** <Aromatherapy Usage, advises, Burnout Prevention><Aromatherapy Usage, connectedTo, Alternative Therapies><Aromatherapy Usage, partOf,

Resilience Building><Alternative Therapies, connectedTo, Aromatherapy Usage><Alternative Therapies, partOf, Resilience Building><Behavior Change Support, connectedTo, Personal Growth Planning><Behavior Change Support, connectedTo, Physical Activity Planning><Burnout Prevention, connectedTo, Detox Programs><Burnout Prevention, partOf, Resilience Building><Detox Programs, connectedTo, Burnout Prevention><Detox Programs, partOf, Resilience Building><Employee Wellness Programs, complies, Behavior Change Support><Employee Wellness Programs, eats, Health Coaching><Environmental Wellness, connectedTo, Holistic Health Assessment><Environmental Wellness, connectedTo, Mindfulness Practice><Environmental Wellness, connectedTo, Physical Activity Planning><Environmental Wellness, sleeps, Physical Activity Planning><Health Coaching, connectedTo, Detox Programs><Health Coaching, connectedTo, Environmental Wellness><Health Coaching, connectedTo, Personal Growth Planning><Health Coaching, eats, Employee Wellness Programs><Holistic Health Assessment, recommends, Personal Growth Planning><Journaling for Wellness, connectedTo, Mindfulness Practice><Journaling for Wellness, connectedTo, Personal Growth Planning><Journaling for Wellness, connectedTo, Physical Activity Planning><Journaling for Wellness, enhances, Massage Therapy><Journaling for Wellness, optimizes, Massage Therapy><Journaling for Wellness, stress-reduces, Environmental Wellness><Mindfulness Practice, connectedTo, Environmental Wellness><Mindfulness Practice, educates, Spectroscopy><Massage Therapy, connectedTo, Personal Growth Planning><Massage Therapy, engages, Health Coaching><Massage Therapy, meditates, Environmental Wellness><Massage Therapy, mindfulness, Personal Growth Planning><Massage Therapy, personalizes, Alternative Therapies><Personal Growth Planning, connectedTo, Behavior Change Support><Personal Growth Planning, connectedTo, Holistic Health Assessment><Personal Growth Planning, connectedTo, Physical Activity Planning><Personal Growth Planning, recommends, Holistic Health Assessment><Physical Activity Planning, connectedTo, Environmental Wellness><Physical Activity Planning, researches, Holistic Health Assessment><Physical Activity Planning, sleeps, Environmental Wellness>"

**Error Analysis:**
Qwen2.5-VL-7B exhibits inconsistent predictions across different renderers for the same underlying topic. For example, in the PlantUML rendering, the gold-standard graph emphasizes role-task relations such as ⟨Detox Programs, eats, Gut Health for Wellness⟩ and ⟨Social Connection Enhancement, protects, Holistic Health Assessment⟩. However, the model output drifts into spurious links like ⟨Resilience Building, complies, Quantum Computing Integration⟩, which are entirely unrelated. In contrast, when the same topic is rendered with Mermaid, the gold standard encodes alternative therapy structures (e.g., ⟨Aromatherapy Usage, advises, Burnout Prevention⟩, ⟨Alternative Therapies, partOf, Resilience Building⟩), yet the model prediction introduces inconsistent or shallow associations, failing to capture the intended containment and functional dependencies. These examples illustrate a clear cross-renderer sensitivity: even for the same semantic topic, the model's outputs diverge substantially depending on the renderer, resulting in structurally mismatched and semantically incoherent graphs.

### K.2 COMPLEX NESTING ERROR ON FLOWGEN TEST SET

> **An Example of Complex Nesting Error on FlowGen Test Set**
>
> 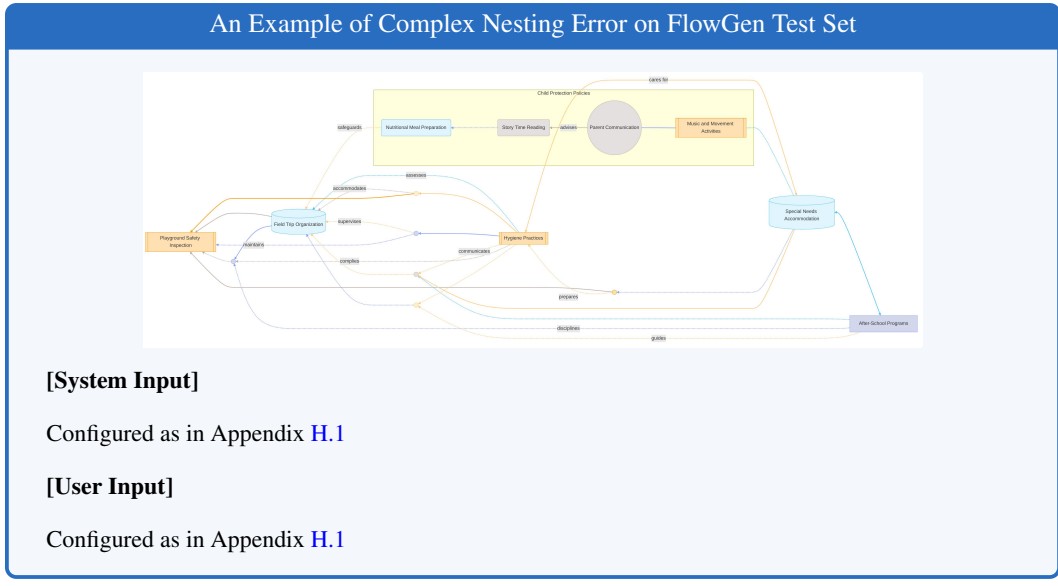
>
> **[System Input]**
>
> Configured as in Appendix H.1
>
> **[User Input]**
>
> Configured as in Appendix H.1

**Qwen2.5-VL-7B Output:**
<After-School Programs, connectedTo, Special Needs Accommodation>
<After-School Programs, disciplines, Field Trip Organization>
<After-School Programs, guides, Hygiene Practices>
<Field Trip Organization, connectedTo, Playground Safety Inspection>
<Field Trip Organization, maintains, Playground Safety Inspection>
<Field Trip Organization, safeguards, Nutritional Meal Preparation>
<Hygiene Practices, assesses, Nutritional Meal Preparation>
<Hygiene Practices, communicates, Field Trip Organization>
<Hygiene Practices, connectedTo, Field Trip Organization>
<Hygiene Practices, connectedTo, Playground Safety Inspection>
<Hygiene Practices, supervises, Field Trip Organization>
<Music and Movement Activities, connectedTo, Special Needs Accommodation>
<Music and Movement Activities, partOf, Child Protection Policies>
<Nutritional Meal Preparation, connectedTo, Field Trip Organization>
<Nutritional Meal Preparation, partOf, Child Protection Policies>
<Parent Communication, advises, Story Time Reading>
<Parent Communication, connectedTo, Music and Movement Activities>
<Parent Communication, partOf, Child Protection Policies>
<Playground Safety Inspection, connectedTo, Field Trip Organization>
<Special Needs Accommodation, cares for, Hygiene Practices>
<Special Needs Accommodation, connectedTo, Music and Movement Activities>
<Special Needs Accommodation, connectedTo, Playground Safety Inspection>
<Special Needs Accommodation, prepares, Field Trip Organization>
<Story Time Reading, advises, Parent Communication>
<Story Time Reading, partOf, Child Protection Policies>
<Transmission Electron Microscopy, connectedTo, Nanodevice Fabrication>
<Transmission Electron Microscopy, connectedTo, Solar Cell Efficiency Enhancement>

**Error Analysis:**
Qwen2.5-VL-7B fails to capture hierarchical containment and role-task dependencies present in the gold standard. While the gold-standard graph encodes explicit *partOf* relations—e.g., situating "Music and Movement Activities", "Parent Communication", "Story Time Reading", and "Nutritional Meal Preparation" under "Child Protection Policies", the model's prediction, however, flattens or distorts these structures: it omits key containment relations, incorrectly shifts responsibilities (e.g., assigning ⟨Nutritional Meal Preparation, partOf, Child Protection Policies⟩ while failing to preserve its other nested *partOf* relations), and even hallucinates irrelevant nodes such as Transmission Electron Microscopy, demonstrating difficulty in preserving complex hierarchies across multiple functional domains.

## K.3  EDGE AMBIGUITY ERROR ON FLOWGEN TEST SET



### An Example of Edge Ambiguity Error on FlowGen Test Set

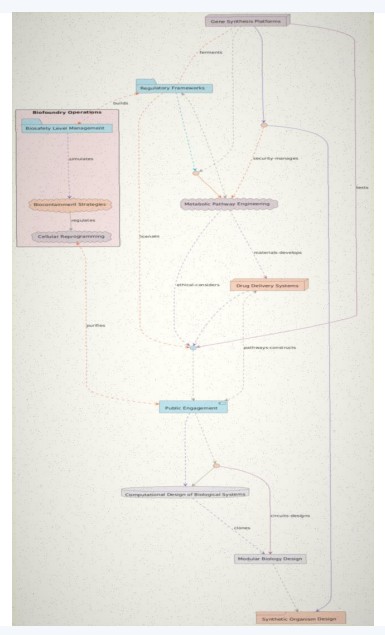

**[System Input]**

Configured as in Appendix H.1

**[User Input]**

Configured as in Appendix H.1



**Qwen2.5-VL-7B Output:**
<Biocontainment Strategies, partOf, Biofoundry Operations>
<Biocontainment Strategies, regulates, Cellular Reprogramming>
<Biosafety Level Management, builds, Regulatory Frameworks>
<Biosafety Level Management, partOf, Biofoundry Operations>
<Biocontainment Strategies, simulates, Biosafety Level Management>
<Cellular Reprogramming, partOf, Biofoundry Operations>
<Cellular Reprogramming, purifies, Public Engagement>
<Computational Design of Biological Systems, clones, Modular Biology Design>
<Computational Design of Biological Systems, connectedTo, Public Engagement>
<Drug Delivery Systems, circuits-designs, Synthetic Organism Design>
<Drug Delivery Systems, materials-develops, Metabolic Pathway Engineering>
<Gene Synthesis Platforms, connectedTo, Metabolic Pathway Engineering>
<Gene Synthesis Platforms, ferments, Regulatory Frameworks>
<Gene Synthesis Platforms, security-manages, Metabolic Pathway Engineering>
<Gene Synthesis Platforms, tests, Synthetic Organism Design>
<Metabolic Pathway Engineering, connectedTo, Drug Delivery Systems>
<Metabolic Pathway Engineering, ethical-considers, Public Engagement>
<Metabolic Pathway Engineering, materials-develops, Drug Delivery Systems>
<Modular Biology Design, connectedTo, Synthetic Organism Design>
<Public Engagement, connectedTo, Computational Design of Biological Systems>
<Public Engagement, connectedTo, Synthetic Organism Design>
<Public Engagement, purifies, Cellular Reprogramming>
<Regulatory Frameworks, builds, Biosafety Level Management>
<Regulatory Frameworks, connectedTo, Metabolic Pathway Engineering>
<Regulatory Frameworks, licenses, Public Engagement>
<Transmission Electron Microscopy, connectedTo, Nanodevice Fabrication>

> **Error Analysis:**
> Qwen2.5-VL-7B's prediction exhibits clear *edge ambiguity* errors, where the directions of edges are frequently misrepresented. In the gold standard, edges encode regulatory or constructive relations (e.g., ⟨Biosafety Level Management, simulates, Biocontainment Strategies⟩), while the model frequently reverses them (e.g., ⟨Biocontainment Strategies, simulates, Biosafety Level Management⟩), inverting the causal logic. In addition, spurious edges and nodes are introduced (e.g., ⟨Transmission Electron Microscopy, connectedTo, Nanodevice Fabrication⟩), which do not exist in the input diagram. As a result, the predicted graph fails to preserve the structural consistency of the scanned flowchart, producing ambiguous and logically inconsistent edge relations.

## K.4   OCR-DRIVEN ERROR ON FLOWGEN TEST SET

**An Example of OCR-Driven Error on FlowGen Test Set**

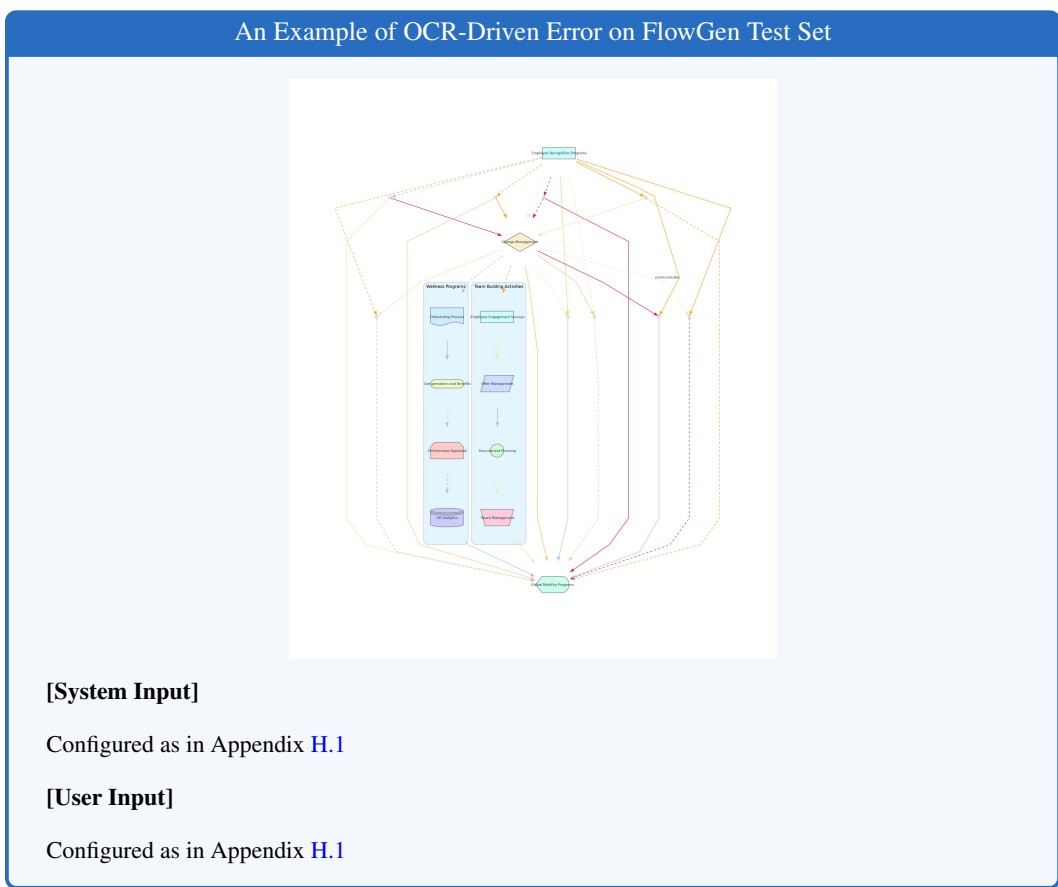

**[System Input]**

Configured as in Appendix H.1

**[User Input]**

Configured as in Appendix H.1

**Qwen2.5-VL-7B Output:**
<Employee Engagement Surveys, connectedTo, Offer Management>
<Employee Recognition Programs, connectedTo, Change Management>
<Employee Recognition Programs, connectedTo, Global Mobility Programs>
<Employee Recognition Programs, connectedTo, Nanodevice Fabrication>
<Employee Recognition Programs, connectedTo, Talent Management>
<HR Analytics, connectedTo, Performance Appraisal>
<Offer Management, connectedTo, Recruitment Planning>
<Performance Appraisal, connectedTo, HR Analytics>
<Recruitment Planning, connectedTo, Talent Management>
<Change Management, connectedTo, Employee Engagement Surveys>
<Change Management, connectedTo, Global Mobility Programs>
<Change Management, connectedTo, Nanodevice Fabrication>
<Global Mobility Programs, connectedTo, Onboarding Process>
<Onboarding Process, connectedTo, Compensation and Benefits>
<Talent Management, connectedTo, Global Mobility Programs>

**Error Analysis:**
Qwen2.5-VL-7B's output reveals typical OCR-driven errors: misrecognized text leads to hallucinated nodes such as "Nanodevice Fabrication", which do not exist in the gold standard. As a result, the model generates spurious edges (e.g., ⟨Employee Recognition Programs, connectedTo, Nanodevice Fabrication⟩) while omitting valid dependencies like ⟨HR Analytics, connectedTo, Global Mobility Programs⟩. These OCR-induced distortions corrupt both the semantics and the topology of the predicted graph, illustrating how recognition noise directly propagates into structural parsing errors.

