# OpenReview forum: "FlowGen: Synthesizing Diverse Flowcharts to Enhance and Benchmark MLLM Reasoning"
_ICLR.cc/2026/Conference — ICLR 2026 Poster_

### Official Review · Reviewer_TxLZ · 2025-10-26

**Soundness:** 2
**Presentation:** 2
**Contribution:** 2
**Rating:** 4
**Confidence:** 4

**Summary:**

This paper introduces FlowGen, a controllable synthesizer designed to generate diverse and structured flowcharts for training and evaluating multimodal large language models (MLLMs). The authors conduct comprehensive experiments showing that fine-tuning on FlowGen significantly improves flowchart parsing and question answering, while also exposing weaknesses in current MLLMs under complex or cross-renderer conditions. However, some aspects of the dataset design and evaluation pipeline need further clarification and empirical support.

**Strengths:**

The paper addresses an important gap in multimodal reasoning benchmarks by introducing a controllable and scalable flowchart synthesis framework, which allows explicit control over graph complexity and rendering diversity.

The proposed FlowGen synthesizer is technically sound and provides a reproducible pipeline for generating both training and testing data, with adjustable parameters such as graph order, branching patterns, and rendering style.

The experiments are comprehensive, covering multiple flowchart-related tasks and a variety of open-source and proprietary MLLMs, which strengthens the empirical validity of the results.

**Weaknesses:**

Weaknesses and Questions

The semantic content of the synthesized flowcharts is said to be drawn from 120 predefined application domains, each providing 40 node names and 40 edge labels. However, the paper does not explain the distribution and diversity of these domains, nor does it provide examples of the node and edge vocabularies. A more detailed breakdown would help readers assess how representative and balanced these semantic domains are. Additionally, the process of generating content “with GPT-4o and refined through human verification” is mentioned but not described in sufficient detail—clarifying this step would strengthen the reproducibility and credibility of the dataset.

While the experiments focus on several flowchart-specific tasks, flowchart understanding is arguably a fundamental multimodal reasoning skill. It would be valuable to evaluate whether models fine-tuned on FlowGen also exhibit transfer improvements on broader benchmarks such as MathVista or MMMU, to demonstrate that FlowGen enhances general reasoning rather than only task-specific performance.

In Section 3.2, the authors mention synthesizing a training set of 11,520 flowcharts but do not explain how the task components of each sample are structured. Are all samples associated with parsing tasks, or do they also include QA tasks or other forms of supervision? Beyond Table 2, more detailed statistics, such as the task type distribution and average tokens per sample, would help readers better understand the dataset’s design and scope.

The evaluation setup involves prompting models with triplets extracted by Qwen2.5-VL-7B fine-tuned on FlowGen data. This design is not fully end-to-end, and may introduce error propagation from the extraction stage. A cleaner approach might be to evaluate each model’s direct performance on FlowGen-derived tasks (e.g., QA) using its own fine-tuned checkpoint, without intermediate feature extraction.

The paper also introduces a synthetic test set generated using the same pipeline. However, it is unclear whether the test set underwent any manual inspection or quality assurance. Since the same generation process is used for both train and test data, some discussion of data overlap prevention and human verification would be helpful to ensure a fair and meaningful evaluation.

Table 6 shows that models trained on specific subsets (e.g., Graph Easy, Medium, Hard) achieve their best results on the corresponding test subsets. This suggests possible limited cross-subset generalization. Have the authors tested a model trained on the entire training set (combining all difficulty levels) to examine whether it performs more robustly across test subsets? Such results would provide a clearer picture of the model’s generalization ability.

**Questions:**

Please see weaknesses

---

> ### Author Response · Authors · 2025-11-21
> **Author response (1/4)**
>
> We sincerely thank you for taking the time to review our manuscript and for your valuable feedback, which **primarily focuses on requests for clarifications and additional experiments**. Below, we address the points you have raised, and these responses will be incorporated into the revised version of the paper.
>
> > **(W1)** The semantic content of the synthesized flowcharts is said to be drawn from 120 predefined application domains, each providing 40 node names and 40 edge labels. However, the paper does not explain the distribution and diversity of these domains, nor does it provide examples of the node and edge vocabularies. A more detailed breakdown would help readers assess how representative and balanced these semantic domains are. Additionally, the process of generating content “with GPT-4o and refined through human verification” is mentioned but not described in sufficient detail—clarifying this step would strengthen the reproducibility and credibility of the dataset.
>
> Thank you for this question. **We will incorporate the following explanation into the paper.**
>
> As outlined in our Prompt for Topic and Node/Edge Names Generation in Appendix D, we first used GPT-4o to produce a pool of candidate topics related to real-world flowchart scenarios. These candidates were manually filtered to remove inappropriate entries and consolidate similar ones. Throughout this process, we drew on a variety of publicly available sources---including industry documentation, course syllabi, and domain-specific datasets---to minimize redundancy, ensure terminological accuracy, and maintain broad domain diversity. This resulted in a final set of 120 distinct topics, each accompanied by its own vocabulary of node names and edge labels. The topics span diverse areas such as business operations, scientific processes, industrial engineering, software systems, healthcare, finance, education, and public services, ensuring wide coverage across multiple fields.
>
> Below we present several sampled topics. For two selected topics, Finance and Healthcare, the corresponding node names and edge labels are illustrated. The full lists are available in `/Synthesizer/my_dictionary.py` in our anonymous GitHub repository.
>
> ```python
> ['Finance', 'Healthcare', 'Manufacturing', 'Education', 'E_commerce', 'Transportation', 'Energy', 'Legal', 'Logistics',  'Telecom', 'Agriculture','Tourism', 'Real Estate','Public Safety','Entertainment',  'Environment','Government', 'Retail', 'Automotive', 'Aerospace',  'Insurance',  'Cybersecurity', 'Smart Home', 'Smart City', 'Biotechnology', 'Pharmaceuticals', 'Robotics', 'Construction', 'Media', 'Sports', 'Fashion', 'Human Resources', 'Recruitment','Customer Service', 'Supply Chain', 'Mining','Maritime', 'Space Exploration', 'Climate Science', 'Meteorology','Waste Management', 'Water Management', 'Recycling',  'Forestry', 'Veterinary', 'Childcare',  'Elderly Care', 'Disaster Management', 'Military',   'Aviation', 'Navigation', '3D Printing', 'Nanotechnology', 'Quantum Computing',  'AR or VR', 'Gaming', 'Music',   'Film Production', 'Content Creation', ...]
>
> NODE_NAMES = {
>     'Finance': [
>         'Account Verification', 'Credit Scoring', 'Approval Decision', 'Fraud Check', 'Transaction Audit', ...],
>     'Healthcare': [
>         'Patient Intake', 'Lab Test', 'Diagnosis', 'Prescription', 'Medical History', ...],
>     ...}
>
> EDGE_LABELS = {
>     'Finance': [
>         'verifies', 'approves', 'rejects', 'checks', 'calculates', 'recommends', ...],
>     'Healthcare': [
>         'diagnoses', 'examines', 'tests', 'treats', 'scans',  'monitors', ...]
>     ...}
> ```
>
> > **(W2)** While the experiments focus on several flowchart-specific tasks, flowchart understanding is arguably a fundamental multimodal reasoning skill. It would be valuable to evaluate whether models fine-tuned on FlowGen also exhibit transfer improvements on broader benchmarks such as MathVista or MMMU, to demonstrate that FlowGen enhances general reasoning rather than only task-specific performance.
>
> Thank you for this valuable feedback. In response to your suggestion, **we have initiated additional experiments. (Update: Results are now available in our 4th response.)** However, we observed that fine-tuning models exclusively on data generated by FlowGen leads to overfitting on flowchart parsing tasks---as might be expected---significantly impairing their ability to perform other types of tasks. To address this issue, and following common practice, we plan to incorporate training data from additional tasks to preserve the model's general capabilities. This adjustment requires additional effort, and we aim to complete these experiments before the end of the discussion period.

---

> ### Author Response · Authors · 2025-11-21
> **Author response (2/4)**
>
> > **(W3)** In Section 3.2, the authors mention synthesizing a training set of 11,520 flowcharts but do not explain how the task components of each sample are structured. Are all samples associated with parsing tasks, or do they also include QA tasks or other forms of supervision? Beyond Table 2, more detailed statistics, such as the task type distribution and average tokens per sample, would help readers better understand the dataset’s design and scope.
>
> Thank you for this question. FlowGen is specifically designed for flowchart parsing, and accordingly, **all synthesized samples are dedicated solely to parsing tasks.** Each sample consists of a flowchart image paired with its corresponding triple-based graph representation. No additional forms of supervision---such as QA labels or other task types---are included in the current dataset. That said, your suggestion inspires us to consider incorporating such supervision signals in future extensions to support a broader range of downstream applications.
>
> > **(W4)** The evaluation setup involves prompting models with triplets extracted by Qwen2.5-VL-7B fine-tuned on FlowGen data. This design is not fully end-to-end, and may introduce error propagation from the extraction stage. A cleaner approach might be to evaluate each model’s direct performance on FlowGen-derived tasks (e.g., QA) using its own fine-tuned checkpoint, without intermediate feature extraction.
>
> Thank you for suggesting this potentially more effective approach for utilizing our synthetic data. Similar to our response to W2, **we have initiated additional experiments** and will report the results once available. **(Update: Results are now available in our 4th response.)** It is worth noting, however, that even if some degree of error propagation exists in the current extraction process, the extracted triples still contribute to a significant improvement in QA accuracy, as demonstrated in Table 4 in the paper. This already substantiates the value of our work.
>
> > **(W5)** The paper also introduces a synthetic test set generated using the same pipeline. However, it is unclear whether the test set underwent any manual inspection or quality assurance.
>
> We did not perform manual inspection or additional quality assurance on the synthetic test set, as the synthesis process itself---rendering randomly constructed flowchart graphs---is designed to be deterministic and **inherently free from common data generation errors**.
>
> > Since the same generation process is used for both train and test data, some discussion of data overlap prevention and human verification would be helpful to ensure a fair and meaningful evaluation.
>
> We appreciate the opportunity to clarify this point, which appears to have been unclear in our original description. We would like to clarify that FlowGen-generated flowcharts are designed to be used **either** as training data (as in Section 3) **or** as test data (as in Section 4), **but not for both in the same experiment**, in order to prevent potential biases.
>
> In the specific experiment corresponding to Table 6, we intentionally used FlowGen-generated data for both training and testing in order to stress-test generalization capability. The results show that **even when models could potentially exploit biases in the data, they still struggled on our test set**. This further suggests that the difficulty of our benchmark stems from the complex topological structures---such as branching, nesting, and diverse layouts---which are not easily learnable through superficial patterns.
>
> We will incorporate this clarification in the paper.

---

> ### Author Response · Authors · 2025-11-21
> **Author response (3/4)**
>
> > **(W6)** Table 6 shows that models trained on specific subsets (e.g., Graph Easy, Medium, Hard) achieve their best results on the corresponding test subsets. This suggests possible limited cross-subset generalization. Have the authors tested a model trained on the entire training set (combining all difficulty levels) to examine whether it performs more robustly across test subsets? Such results would provide a clearer picture of the model’s generalization ability.
>
> Thank you for this constructive suggestion. Following your advice, **we conducted additional experiments and obtained expected results.**
>
> Specifically, we fine-tuned models on a combined training set comprising synthetic data across all six difficulty levels from Table 6. As shown in the extended table below, these models indeed demonstrate more robust performance across test subsets. For Qwen2.5-VL-3B and MiniCPM-V2.6-8B, fine-tuning on the combined dataset yields the best results on some or all test subsets. For Qwen2.5-VL-7B, it also exhibits consistent robustness across different difficulty levels. We will incorporate these findings into the revised version of the paper.
>
> | **Model**                | **Graph-Easy** | **Graph-Medium** | **Graph-Hard** | **Scanned-Easy** | **Scanned-Medium** | **Scanned-Hard** |
> | ------------------------ | -------------- | ---------------- | -------------- | ---------------- | ------------------ | ---------------- |
> | **Qwen2.5-VL-3B**            | 20.4           | 6.2              | 2.9            | 10.2             | 10.0               | 9.0              |
> | +SFT on Graph Easy   | **50.2**       | 24.0             | 16.0           | 33.0             | 30.5               | 29.0             |
> | +SFT on Graph Medium | 36.3           | **32.9**         | 28.2           | 33.1             | 33.1               | 31.3             |
> | +SFT on Graph Hard   | 26.9           | 30.5             | **30.3**       | 29.4             | 29.8               | 28.4             |
> | +SFT on Scanned Easy     | 45.1           | 31.8             | 29.0           | 34.8             | 35.3               | 34.1             |
> | +SFT on Scanned Medium   | 43.6           | 32.2             | 28.5           | 34.8             | 35.5               | 33.8             |
> | +SFT on Scanned Hard     | 43.7           | 32.6             | 29.3           | 35.6             | 35.5               | **35.0**         |
> | +SFT on Combined     | 49.9           | 31.0             | 24.6           | **36.3**         | **35.7**           | 33.1             |
> | **Qwen2.5-VL-7B**            | 35.7           | 10.9             | 7.7            | 18.9             | 18.5               | 17.2             |
> | +SFT on Graph Easy   | **74.9**       | 35.0             | 21.2           | 45.3             | 44.3               | 41.9             |
> | +SFT on Graph Medium | 67.3           | **52.4**         | **44.0**       | **56.2**         | **55.3**           | **52.3**         |
> | +SFT on Graph Hard   | 39.1           | 43.6             | 43.3           | 42.6             | 42.9               | 40.8             |
> | +SFT on Scanned Easy     | 66.8           | 47.8             | 41.6           | 53.9             | 53.1               | 49.7             |
> | +SFT on Scanned Medium   | 68.8           | 48.1             | 42.0           | 54.4             | 53.6               | 51.0             |
> | +SFT on Scanned Hard     | 66.0           | 47.1             | 42.0           | 53.0             | 52.6               | 50.2             |
> | +SFT on Combined     | 66.5           | 47.2             | 42.6           | 52.8             | 51.7               | 50.2             |
> | **MiniCPM-V2.6-8B**          | 7.3            | 2.1              | 2.1            | 4.0              | 4.3                | 3.5              |
> | +SFT on Graph Easy       | 33.0           | 13.1             | 9.0            | 18.8             | 18.1               | 17.7             |
> | +SFT on Graph Medium     | 23.9           | 19.6             | 15.2           | 20.0             | 19.9               | 18.1             |
> | +SFT on Graph Hard       | 17.5           | 17.4             | 17.4           | 17.8             | 17.9               | 16.8             |
> | +SFT on Scanned Easy     | 29.6           | 19.1             | 16.5           | 22.2             | 22.1               | 20.8             |
> | +SFT on Scanned Medium   | 29.6           | 19.0             | 16.1           | 22.3             | 22.3               | 20.4             |
> | +SFT on Scanned Hard     | 27.1           | 18.0             | 16.3           | 21.3             | 20.4               | 19.4             |
> | +SFT on Combined | **39.5**       | **24.2**         | **20.6**       | **29.0**         | **28.8**           | **27.1**         |

---

> ### Author Response · Authors · 2025-11-24
> **Author response (4/4)**
>
> Thank you for your patience. We have now completed the two remaining requested experiments and are pleased to present the results below.
>
> > **(W2)** ... to evaluate whether models fine-tuned on FlowGen also exhibit transfer improvements on broader benchmarks such as MathVista or MMMU ...
>
> Following the reviewer's request, **we further investigated whether FlowGen fine-tuning transfers to broader multimodal reasoning.**
>
> Specifically, following the methodology of Dong et al. (2024) [1], who demonstrated that naively mixing heterogeneous tasks in multi-task supervised fine-tuning (SFT) often leads to performance conflicts, while sequential training tends to cause catastrophic forgetting of previously learned skills, we adopted an *interleaved training strategy*. Under this approach, flowchart-parsing samples from FlowGen and question-answering samples from MMMU are mixed at the batch level, enabling the model to jointly optimize both capabilities without compromising either.
>
> [1] Dong, et al. How abilities in large language models are affected by supervised fine-tuning data composition. ACL 2024.
>
> Since MMMU does not provide a standard training split, we used its 900 validation samples for SFT and its 150 development samples for testing. Specifically, we fine-tuned each model under two settings:
> 1. using only MMMU data ("+SFT on MMMU"), or
> 2. using a size-matched mixed dataset in which half of the MMMU samples were replaced with FlowGen triplet-parsing data ("+SFT on Mixed").
>
> The total number of training samples was kept identical between the two settings, ensuring that any performance improvement can be attributed to the inclusion of FlowGen data rather than an increase in training volume.
>
> As shown in the table below, **models enhanced with FlowGen consistently improve on MMMU**, a benchmark spanning diverse general-domain reasoning tasks unrelated to flowcharts. This indicates that FlowGen strengthens fundamental structural and spatial reasoning abilities, which in turn support performance beyond task-specific flowchart understanding.
>
> | Model           |   MMMU   |
> | :-------------- | :------: |
> | Qwen2.5-VL-3B   |   53.3   |
> | + SFT on MMMU    |   59.3   |
> | + SFT on Mixed   | **60.2** |
> | Qwen2.5-VL-7B   |   58.6   |
> | + SFT on MMMU    |   62.6   |
> | + SFT on Mixed   | **65.3** |
> | MiniCPM-V2.6-8B |   49.4   |
> | + SFT on MMMU    |   53.3   |
> | + SFT on Mixed   | **54.0** |
>
> > **(W4)** ... to evaluate each model's direct performance on FlowGen-derived tasks (e.g., QA) using its own fine-tuned checkpoint, without intermediate feature extraction.
>
> Following the reviewer's request, **we conducted a fully end-to-end evaluation using each model’s own fine-tuned checkpoint, without intermediate feature extraction.**
>
> Specifically, on the FlowVQA and FlowLearn benchmarks, we fine-tuned each model under two settings:
> 1. using its original QA training set ("SFT on its original training set"), or
> 2. using a size-matched mixed dataset in which half of the original training samples were replaced with FlowGen triplet-parsing data ("SFT on mixed").
>
> The total number of training samples was kept the same in both settings, ensuring that any performance gain cannot be attributed to increased data volume.
>
> As shown in the table below, **end-to-end accuracy is observed to improve with FlowGen for most models and datasets,** indicating that FlowGen generally strengthens models' intrinsic ability to parse graph structures and spatial relationships. These results complement our triplet-based evaluation by demonstrating that FlowGen enhances performance even when models are evaluated directly on downstream tasks using their own fine-tuned checkpoints---validating that FlowGen delivers authentic end-to-end performance gains.
>
> | Model                          | FlowVQA  | FlowLearn |
> | :----------------------------- | :------: | :-------: |
> | Qwen2.5-VL-3B                  |   64.6   |   72.3    |
> | + SFT on its original training set |   81.3   | **85.1**  |
> | + SFT on mixed                 | **85.6** |   84.7    |
> | Qwen2.5-VL-7B                  |   74.6   |   71.1    |
> | + SFT on its original training set |   88.2   |   85.3    |
> | + SFT on mixed                 | **89.2** | **89.1**  |
> | MiniCPM-V2.6-8B                |   61.6   |   80.9    |
> | + SFT on its original training set |   73.2   |   86.2    |
> | + SFT on mixed                 | **77.6** | **87.5**  |

---

> ### Author Response · Authors · 2025-11-27
> **A gentle reminder**
>
> Dear Reviewer,
>
> As the discussion period is drawing to a close, we wanted to kindly follow up on our responses. We sincerely hope they have addressed your comments satisfactorily, and we would be happy to provide any additional information or clarification if necessary.
>
> Thank you very much for your time and consideration.
>
> Best regards,
>
> The Authors

---

### Official Review · Reviewer_3UeF · 2025-10-27

**Soundness:** 2
**Presentation:** 3
**Contribution:** 2
**Rating:** 2
**Confidence:** 5

**Summary:**

This paper presents FlowGen, a controllable flowchart synthesizer for training and benchmarking multimodal LLMs. The generator exposes seven knobs (graph order, split/merge arrows, branching factor, density, unlabeled-edge ratio, nested subgraphs) and renders with Mermaid, Graphviz, PlantUML, and Diagrams to induce style diversity (Fig. 2, p.3). Using FlowGen, the authors build a 11,520‑image training set (Table 2, p.6) and a six‑subset test suite spanning structural difficulty and scanned‑image degradation. Across six parsing benchmarks, fine‑tuning open‑source MLLMs on FlowGen yields large F1 gains (e.g., Qwen2.5‑VL‑7B: FlowLearn 43.2→60.1 strict F1; hdBPMN 16.6→18.3; Table 3, p.6), and supplying triplets extracted by a FlowGen‑tuned model improves flowchart QA accuracy for multiple models (Table 4, p.7). A new FlowGen test set exposes low base performance for both open‑source and proprietary models (e.g., GPT‑4o ≤25% F1 on most subsets) and shows that FlowGen‑SFT substantially helps (Table 6, p.9). Ablations indicate multi‑renderer training and explicit modeling of nesting/split‑merge structures matter (Table 5, p.7).

**Strengths:**

- Combines controllable graph parameters with multi‑renderer outputs; introduces scanned‑style perturbations and a matched train/test pipeline
- Extensive, multi‑model evaluation; clear ablations demonstrating the role of renderer diversity, nesting, and split/merge
- Well‑specified metrics and prompts (Appendix D) and transparent training setup.
- Offers a scalable path to stress‑test MLLM reasoning over structured diagrams; highlights consistent weaknesses

**Weaknesses:**

1. Evaluation Design Masks True Model Capabilities

Table 4's approach—feeding triplets extracted by a FlowGen-tuned Qwen-7B to all models—introduces two fundamental biases. First, it conflates extraction ability with reasoning capacity, making it impossible to isolate genuine question-answering performance. Second, it privileges models that happen to align with the chosen extractor's error profile. The paper should instead report: (i) QA performance without triplets as a baseline, (ii) QA with gold-standard triplets to measure pure reasoning, and (iii) QA using each model's self-extracted triplets to properly separate parsing from comprehension.

2. Virtual Node Encoding Introduces Unvalidated Structural Bias

The split/merge virtual-node representation deviates from standard BPMN conventions (gateways, arrow semantics) in ways that could fundamentally bias what models learn. While the authors acknowledge this as future work (Conclusion, p.9), no quantitative analysis compares this encoding against alternatives. Without understanding how much this choice shapes results, we cannot assess whether findings generalize to real-world diagram understanding.

3. Underwhelming Performance on Real Data Raises Generalization Questions

The best FlowGen-SFT result on hdBPMN remains modest—just 18.3% F1 for Qwen-7B (Table 3). This gap between synthetic training performance and real-world evaluation suggests the approach may not transfer beyond stylized, template-generated data. The paper needs substantially larger, truly out-of-distribution real-world test sets to validate practical applicability.

4. Train-Test Splits Are Distribution-Matched, Not Truly Adversarial

FlowGen's test sets are disjoint from training but deliberately distribution-matched (Sec. 4.1.1, p.8). This design choice likely inflates apparent robustness compared to unconstrained deployment scenarios. Cross-renderer evaluations (where training and test use different rendering engines) and unseen-topic tests would provide more realistic difficulty estimates.

5. "Scanned" Augmentations Are Synthetic, Not Ecological

Table 10's scanned-style effects (p.20) come from Pillow/OpenCV filters, not actual scanning or photography. Real documents exhibit sensor noise, lighting variation, physical degradation, and artifacts that synthetic perturbations cannot fully capture. Evaluation on genuinely scanned, photographed, and hand-drawn diagrams is essential for validating real-world robustness.

6. Vocabulary Verification Lacks Transparency

Node/edge vocabularies are generated by GPT-4o then "human-verified" (Sec. 2.2.2, p.4), but verification coverage and inter-annotator reliability are not reported. Without knowing what percentage was checked, by how many annotators, and with what agreement rates, we cannot assess label quality or rule out systematic template artifacts.

7. "Unlimited Diversity" Claim Is Unsupported

The assertion that FlowGen generates unlimited diverse flowcharts contradicts the method's reliance on 120 static domain dictionaries with fixed vocabularies. Examination reveals repetitive structures, especially in "complex" configurations. The system samples from finite pools—it does not model true domain variation or generate genuinely novel structures.

8. Random Semantic Assignment Undermines Reasoning Validity

Labels are randomly sampled from GPT-generated word lists without semantic validation or causal coherence. Many examples show nonsensical flows (e.g., "Health Coaching → Detox Program → Journaling Wellness"). This fundamentally undermines the benchmark's value as a reasoning test: models learn to parse arbitrary structures, not understand real-world logic or causal relationships.

9. Methodologically Inappropriate Baseline Comparisons

Comparing fine-tuned open-source MLLMs to proprietary reasoning-focused models like Gemini-2.5 is not methodologically sound. Gemini is architecturally designed for multi-step reasoning chains, not direct diagram parsing, making performance comparisons misleading. The evaluations conflate architectural differences with actual diagram understanding capability.

10. Missing Critical Semantic Ablation

The central claim—that FlowGen improves visual reasoning—depends on semantically labeled nodes and edges. Yet the paper never evaluates performance when semantic labels are removed or randomized. This omission is critical: observed improvements could stem entirely from textual cues in the image rather than genuine spatial or structural reasoning. Without this ablation, we cannot determine what models actually learn.

11. Limited Conceptual Contribution

Beyond reporting performance numbers, the paper provides little insight into why MLLMs struggle with flowcharts or how they might fundamentally improve. The work essentially demonstrates that synthetic fine-tuning increases scores on similar synthetic data—an expected result that doesn't advance our understanding of diagram comprehension or suggest paths toward more robust visual reasoning.

**Questions:**

- Can you report QA with (a) gold triplets, (b) no triplets, and (c) per‑model self‑extracted triplets? How sensitive are Table 4 gains to extractor accuracy?
- Have you tried ontology mapping, lemmatization, or embedding‑based matching to reduce synonym penalties in triplet evaluation? How do rankings change?
- What is performance when training on one renderer and testing on another (full cross‑matrix), beyond the single‑renderer ablation in Table 5?
- Can you quantify the gap between virtual‑node split/merge and a gateway/edge‑semantics encoding (e.g., BPMN‑style gateways) on hdBPMN?

---

> ### Author Response · Authors · 2025-11-21
> **Author response (1/7)**
>
> We sincerely thank you for taking the time to review our paper. Below, we address the concerns you raised.
>
> >**(W1)** Evaluation Design Masks True Model Capabilities
> >
> >Table 4's approach—feeding triplets extracted by a FlowGen-tuned Qwen-7B to all models—introduces two fundamental biases. First, it conflates extraction ability with reasoning capacity, making it impossible to isolate genuine question-answering performance. Second, it privileges models that happen to align with the chosen extractor's error profile. The paper should instead report: (i) QA performance without triplets as a baseline, (ii) QA with gold-standard triplets to measure pure reasoning, and (iii) QA using each model's self-extracted triplets to properly separate parsing from comprehension.
>
> Following your request, **we have conducted additional experiments, and the results confirm the usefulness of FlowGen.**
>
> Specifically, we implemented the three experimental settings outlined in the review:
> (i) QA **Without Triplets**,
> (ii) QA with **Gold-Standard Triplets**, and
> (iii) QA using each model's **Self-Extracted Triplets**.
>
> The results on two datasets are presented in the following two tables, which extend the original Table 4. Compared to the Without Triplets baseline, models perform nearly identically when using Self-Extracted Triplets---indicating their limited intrinsic ability to parse flowchart structures. As expected, models achieve the strongest performance with Gold-Standard Triplets. Notably, this level of effectiveness is closely approached when using triplets extracted by a model fine-tuned on FlowGen-generated data. This demonstrates that supervision from FlowGen enhances MLLMs' parsing capability, which in turn brings substantial benefits to downstream QA tasks.
>
> We will include these results into the paper.
>
> **Performance comparison on FlowVQA:**
>
> | Model | Without Triplets | Gold-Standard Triplets | Self-Extracted Triplets | FlowGen-Extracted Triplets |
> | :-------------- | :--------: | :-----------: | :---------------------: | :----------: |
> | Qwen2.5-VL-3B   |    64.6    |   **75.6**    |          66.2           |     73.4     |
> | Qwen2.5-VL-7B   |    74.6    |   **83.2**    |          77.1           |     81.1     |
> | MiniCPM-V2.6-8B |    61.6    |   **71.4**    |          61.0           |     69.1     |
>
> **Performance comparison on AI2D:**
>
> | Model | Without Triplets | Gold-Standard Triplets | Self-Extracted Triplets | FlowGen-Extracted Triplets |
> | :-------------- | :--------: | :-----------: | :---------------------: | :----------: |
> | Qwen2.5-VL-3B   |    50.7    |   **53.5**    |          51.2           |     51.4     |
> | Qwen2.5-VL-7B   |    60.9    |   **62.5**    |          60.5           |     61.1     |
> | MiniCPM-V2.6-8B |    32.0    |   **45.0**    |          32.6           |     42.8     |

---

> ### Author Response · Authors · 2025-11-21
> **Author response (2/7)**
>
> >**(W2)** Virtual Node Encoding Introduces Unvalidated Structural Bias
> >
> >The split/merge virtual-node representation deviates from standard BPMN conventions (gateways, arrow semantics) in ways that could fundamentally bias what models learn. While the authors acknowledge this as future work (Conclusion, p.9), no quantitative analysis compares this encoding against alternatives. Without understanding how much this choice shapes results, we cannot assess whether findings generalize to real-world diagram understanding.
>
> We would like to clarify that while BPMN is widely used, it is not the only flowcharting convention employed in practice. The split/merge virtual-node representation adopted in FlowGen is inspired by other commonly used notations---such as unnamed junctions, connector symbols, and implicit merge points---which are prevalent across various diagramming styles.
>
> Following your request, **we have conducted additional experiments, and the results support our design choice.**
>
> Specifically, we replaced the virtual split/merge nodes in our framework with standard BPMN-style gateways (XOR, AND, OR) and reevaluated performance across four datasets. As shown in the following table, this modification led to noticeable performance degradation on datasets containing flowcharts that do not strictly follow BPMN specifications, while offering only marginal gains on hdBPMN. These findings indicate that strictly adhering to BPMN conventions limits cross-style generalization, whereas FlowGen's virtual-node representation offers a more robust and renderer-agnostic structural encoding.
>
> We sincerely appreciate this comment and will consider incorporating greater stylistic diversity in future versions of FlowGen.
>
> | Model                               | FlowVQA  |   CBD    |   FC_B   |  hdBPMN  |
> | :---------------------------------- | :------: | :------: | :------: | :------: |
> | Qwen2.5-VL-3B                       |   12.7   |   17.9   |   8.6    |   2.3    |
> | +SFT (w/ virtual split/merge nodes) | **51.3** | **49.8** | **34.6** |   8.4    |
> | +SFT (w/ BPMN-style gateways)       |   39.7   |   38.5   |   25.9   | **10.5** |
> | Qwen2.5-VL-7B                       |   59.4   |   49.1   |   32.0   |   16.6   |
> | +SFT (w/ virtual split/merge nodes) | **70.9** | **55.4** | **40.8** |   18.3   |
> | +SFT (w/ BPMN-style gateways)       |   63.9   |   49.9   |   31.2   | **19.4** |
> | MiniCPM-V2.6-8B                     |   17.6   |   20.4   |   8.8    |   3.4    |
> | +SFT (w/ virtual split/merge nodes) | **44.0** | **37.9** | **22.9** | **3.6**  |
> | +SFT (w/ BPMN-style gateways)       |   29.2   |   24.6   |   10.8   |   3.5    |

---

> ### Author Response · Authors · 2025-11-21
> **Author response (3/7)**
>
> >**(W3)** Underwhelming Performance on Real Data Raises Generalization Questions
> >
> >The best FlowGen-SFT result on hdBPMN remains modest—just 18.3\% F1 for Qwen-7B (Table 3). This gap between synthetic training performance and real-world evaluation suggests the approach may not transfer beyond stylized, template-generated data. The paper needs substantially larger, truly out-of-distribution real-world test sets to validate practical applicability.
>
> We agree that robust evaluation on real-world data is crucial. In fact, as highlighted in the Introduction section, a key motivation behind developing FlowGen is the scarcity of existing flowchart datasets that adequately capture the diversity and complexity of real-world flowcharts---a limitation clearly illustrated in Table 1 in the paper. **Should the reviewer be aware of any larger, semantically annotated real-world flowchart datasets, we would be very grateful for the reference and would gladly incorporate them in future experiments.**
>
> While performance on hdBPMN remains modest, it is important to note that models trained with FlowGen-SFT **consistently improve on other out-of-distribution real-world datasets**, such as FC_A and FC_B. These datasets consist of non-template, hand-drawn flowcharts, and the observed gains (as shown in Table 3) demonstrate that FlowGen-based training offers meaningful generalization even to challenging and naturally varied flowchart styles.
>
> We also wish to emphasize that hdBPMN presents unique difficulties, including deeply nested structures (illustrated in Appendix E.1) and significant OCR errors caused by irregular handwriting (illustrated in Appendix E.3). These factors partly explain the lower absolute scores. We plan to incorporate greater stylistic and structural diversity in future versions of FlowGen to further improve real-world applicability.
>
> > **(W4)** Train-Test Splits Are Distribution-Matched, Not Truly Adversarial
> >
> >    FlowGen's test sets are disjoint from training but deliberately distribution-matched (Sec. 4.1.1, p.8). This design choice likely inflates apparent robustness compared to unconstrained deployment scenarios. Cross-renderer evaluations (where training and test use different rendering engines) and unseen-topic tests would provide more realistic difficulty estimates.
>
> We appreciate the opportunity to clarify this point, which appears to have been unclear in our original description. We would like to clarify that FlowGen-generated flowcharts are designed to be used **either** as training data (as in Section 3) **or** as test data (as in Section 4), **but not for both in the same experiment**, in order to prevent potential biases.
>
> In the specific experiment corresponding to Table 6, we intentionally used FlowGen-generated data for both training and testing in order to stress-test generalization capability. The results show that **even when models could potentially exploit biases in the data, they still struggled on our test set**. This further suggests that the difficulty of our benchmark stems from the complex topological structures---such as branching, nesting, and diverse layouts---which are not easily learnable through superficial patterns.
>
> We will incorporate this clarification in the paper.
>
> >**(W5)** "Scanned" Augmentations Are Synthetic, Not Ecological
> >
> >Table 10's scanned-style effects (p.20) come from Pillow/OpenCV filters, not actual scanning or photography. Real documents exhibit sensor noise, lighting variation, physical degradation, and artifacts that synthetic perturbations cannot fully capture. Evaluation on genuinely scanned, photographed, and hand-drawn diagrams is essential for validating real-world robustness.
>
> We fully agree that robust evaluation on real-world data is essential. To that end, **we have already conducted experiments on three hand-drawn flowchart datasets**---FC\_A, FC\_B, and hdBPMN---where training with FlowGen-generated data led to substantial performance improvements across various models and datasets, as demonstrated in Table 3 of the paper.
>
> While the scanned-style augmentations used in our study are synthetic, they already pose considerable challenges to current models, as reflected in the results of Table 6. This indicates that our approach represents a meaningful step toward assessing real-world robustness in flowchart parsing. **If the reviewer is aware of any publicly available datasets comprising genuinely scanned or photographed flowcharts, we would greatly appreciate the reference and would be eager to include such data in future experiments.**

---

> ### Author Response · Authors · 2025-11-21
> **Author response (4/7)**
>
> > **(W6)** Vocabulary Verification Lacks Transparency
> >
> > Node/edge vocabularies are generated by GPT-4o then "human-verified" (Sec. 2.2.2, p.4), but verification coverage and inter-annotator reliability are not reported. Without knowing what percentage was checked, by how many annotators, and with what agreement rates, we cannot assess label quality or rule out systematic template artifacts.
>
> Thank you for this question. **We will incorporate the following explanation into the paper.**
>
> As outlined in our Prompt for Topic and Node/Edge Names Generation in Appendix D, we first used GPT-4o to produce a pool of candidate topics related to real-world flowchart scenarios. These candidates were manually filtered to remove inappropriate entries and consolidate similar ones. Throughout this process, we drew on a variety of publicly available sources---including industry documentation, course syllabi, and domain-specific datasets---to minimize redundancy, ensure terminological accuracy, and maintain broad domain diversity. This resulted in a final set of 120 distinct topics, each accompanied by its own vocabulary of node names and edge labels. The topics span diverse areas such as business operations, scientific processes, industrial engineering, software systems, healthcare, finance, education, and public services, ensuring wide coverage across multiple fields. The full vocabulary is available in `/Synthesizer/my_dictionary.py` in our anonymous GitHub repository.
>
> To directly address the reviewer's concern: **all GPT-generated vocabulary entries were manually reviewed and filtered by the authors.** Since the verification was conducted collaboratively by a small team of domain-knowledgeable researchers---with ambiguous or contentious cases resolved through discussion---traditional inter-annotator agreement metrics were not applicable in this setting.
>
> > **(W7)** "Unlimited Diversity" Claim Is Unsupported
> >
> > The assertion that FlowGen generates unlimited diverse flowcharts contradicts the method's reliance on 120 static domain dictionaries with fixed vocabularies. Examination reveals repetitive structures, especially in "complex" configurations. The system samples from finite pools—it does not model true domain variation or generate genuinely novel structures.
>
> The term "unlimited" in Table 1 **refers specifically to the number of flowcharts** that can be synthesized, **not to the uniqueness of their structural or lexical features**. While our current implementation utilizes a finite set of 120 domain dictionaries, the combinatorial space of graph order, arrow types, branching factors, density, edge labels, nesting structures, shapes and colors is vast, enabling the generation of a tremendous number of flowcharts with distinct structures and styles. Furthermore, the framework is designed to be extensible---the dictionaries can be seamlessly expanded to incorporate new domains and vocabularies, thereby continuously enhancing the diversity and real-world coverage of the generated flowcharts.
>
> > **(W8)** Random Semantic Assignment Undermines Reasoning Validity
> >
> > Labels are randomly sampled from GPT-generated word lists without semantic validation or causal coherence. Many examples show nonsensical flows (e.g., "Health Coaching → Detox Program → Journaling Wellness"). This fundamentally undermines the benchmark's value as a reasoning test: models learn to parse arbitrary structures, not understand real-world logic or causal relationships.
>
> As consistently emphasized in our paper, the primary focus of our work is on evaluating and enhancing the ability of models to handle **structural complexity and visual diversity** in flowchart parsing and its downstream applications---**not on causal reasoning**. Accordingly, the use of GPT-generated labels without semantic validation or causal coherence is an intentional design choice, enabling the FlowGen synthesis process to scale efficiently.
>
> The effectiveness of FlowGen as a training resource is clearly demonstrated by the performance improvements reported in Section 3. As a benchmark for testing, it is important to note that our task is strictly focused on flowchart parsing, where **the expected output is a set of structural triplets---not causal inference**. Our experiments and error analysis in Section 4 reveal that current MLLMs still face significant challenges in areas such as cross-renderer generalization, complex nesting, and other structural and visual complexities. These findings affirm the value of FlowGen as a rigorous testbed for evaluating structural reasoning capabilities.

---

> ### Author Response · Authors · 2025-11-22
> **Author response (5/7)**
>
> > **(W9)** Methodologically Inappropriate Baseline Comparisons
> >
> > Comparing fine-tuned open-source MLLMs to proprietary reasoning-focused models like Gemini-2.5 is not methodologically sound. Gemini is architecturally designed for multi-step reasoning chains, not direct diagram parsing, making performance comparisons misleading. The evaluations conflate architectural differences with actual diagram understanding capability.
>
> We acknowledge the reviewer's concern regarding architectural differences between models. Please note that Gemini-2.5 is not the only proprietary model included in our comparison; **we also evaluated against GPT-4o and GLM-4V-Plus.**
>
> While we agree that model architecture influences performance, we included Gemini-2.5 as a baseline for two main reasons:
>
> First, in its technical report, Gemini-2.5 is explicitly evaluated on "Image Understanding" using the MMMU benchmark. This indicates that **the model is not designed purely for text-based reasoning chains, but is also considered relevant for tasks involving visual and structural comprehension**---including diagram parsing, even if not originally tailored for it.
>
> Second, several **recent chart understanding studies have adopted Gemini-2.5 as a baseline for evaluating visual reasoning performance**, such as Tang et al. (2025, arXiv:2505.13444) and Shin et al. (2025, arXiv:2509.18425). This supports its legitimacy as a meaningful reference model in our evaluation, where it serves as one of several---not the sole---baseline for comparison.
>
> > **(W10)** Missing Critical Semantic Ablation
> >
> > The central claim—that FlowGen improves visual reasoning—depends on semantically labeled nodes and edges. Yet the paper never evaluates performance when semantic labels are removed or randomized. This omission is critical: observed improvements could stem entirely from textual cues in the image rather than genuine spatial or structural reasoning. Without this ablation, we cannot determine what models actually learn.
>
> In FlowGen, the semantic content within each flowchart **is already randomized** in the manner noted by the reviewer. Specifically, topics, node names, and edge labels are independently and uniformly sampled from dictionaries during generation, without any contextual or causal coherence. This high degree of textual randomization prevents models from relying on stable semantic cues, ensuring that performance improvements are attributable to structural and spatial reasoning rather than linguistic patterns.
>
> To further isolate the role of structural information, **we have conducted an additional ablation study as suggested:** all node names and edge labels were replaced with meaningless artificial strings. Node names were substituted with random strings of 5–12 characters starting with a capital letter, and edge labels with random strings of 5–12 lowercase characters---preserving only minimal formatting conventions while removing all meaningful semantics.
>
> As shown in the following table, models trained on these fully meaningless labels exhibit only a modest performance drop compared to those trained with randomized though meaningful labels. Importantly, **the performance gains over base models remain consistent and substantial across models and datasets.** These results confirm that the benefits of FlowGen do not stem primarily from textual semantics, but rather from the model's ability to learn structural patterns, topological relationships, and spatial layouts from the flowcharts.
>
> | Model                                 | FlowVQA  |   CBD    |   FC\_B  |
> | :------------------------------------ | :------: | :------: | :------: |
> | Qwen2.5-VL-3B                         |   12.7   |   17.9   |   8.6  |
> | +SFT (with random labels) | 51.3 | 49.8 | 34.6 |
> | +SFT (with meaningless strings)    |   50.9   |   44.1   |   24.1  |
> | Qwen2.5-VL-7B                         |   59.4   |   49.1   |   32.0   |
> | +SFT (with random labels) | 70.9 | 55.4 | 40.8 |
> | +SFT (with meaningless strings)    |   64.5   |   52.9   |   34.8  |
> | MiniCPM-V2.6-8B                       |   17.6   |   20.4   |   8.8    |
> | +SFT (with random labels) | 44.0 | 37.9 | 22.9 |
> | +SFT (with meaningless strings)    |   35.2   |   29.8   |   13.9   |

---

> ### Author Response · Authors · 2025-11-23
> **Author response (6/7)**
>
> > **(W11)** Limited Conceptual Contribution
> >
> > Beyond reporting performance numbers, the paper provides little insight into why MLLMs struggle with flowcharts or how they might fundamentally improve. The work essentially demonstrates that synthetic fine-tuning increases scores on similar synthetic data—an expected result that doesn't advance our understanding of diagram comprehension or suggest paths toward more robust visual reasoning.
>
> We respectfully suggest that **the reviewer may have overlooked our extensive ablation studies and error analysis**, which were designed precisely to uncover why MLLMs struggle with flowchart comprehension and how their capabilities might be systematically improved.
>
> As detailed in Section 3.3.2, we conducted controlled experiments using synthetically generated data under varying constraints to evaluate **which types of training data**---including multi-renderer examples, nested subgraph structures, and split/merge arrows---**contribute most significantly to enhancing model performance**.
>
> Furthermore, in Section 3.3.3, our error analysis **systematically categorizes frequent failure modes**---including complex nesting, edge ambiguity, and OCR-related errors. This taxonomy helps pinpoint specific weaknesses in current models. Additionally, in Section 4.2.2, **we identify cross-renderer generalization as a previously understudied error type that is clearly exposed by the FlowGen benchmark.**
>
> Collectively, these analyses move beyond mere performance reporting to offer concrete, empirically grounded explanations for where and why MLLMs fall short in flowchart parsing---laying a foundation for more robust visual reasoning systems in the future.
>
> > **(Q1)** Can you report QA with (a) gold triplets, (b) no triplets, and (c) per‑model self‑extracted triplets? How sensitive are Table 4 gains to extractor accuracy?
>
> Our response to W1 has addressed this question. We will include the results in the revised version of the paper.
>
> > **(Q2)** Have you tried ontology mapping, lemmatization, or embedding‑based matching to reduce synonym penalties in triplet evaluation? How do rankings change?
>
> We would like to clarify that our reported F1 and relaxed F1 (rF1) metrics already include normalization steps---such as lowercasing and removal of special characters---to minimize penalties from superficial formatting or synonym variations.
>
> Following the reviewer's suggestion, we have now **incorporated embedding-based matching as an additional evaluation metric.** Specifically, we encode each triplet as a complete unit using Sentence-BERT (all-MiniLM-L6-v2) and apply cosine similarity matching with a threshold of 0.85. As shown in the table below, the resulting Emb-F1 scores exhibit nearly identical ranking patterns and magnitudes as the rF1 metric across models and settings. This strong consistency suggests that **further reduction of synonym-related penalties would not materially alter our conclusions.** More importantly, it reinforces the key finding from our error analysis: the dominant source of text-related errors is not synonym mismatch, but **OCR-induced text corruption**---particularly in flowcharts with handwriting-style rendering or complex nested structures.
>
> | Model         |  FlowVQA: F1  |   rF1    |  Emb-F1  |  CBD: F1  |   rF1    |  Emb-F1  |  FC\_A: F1  |   rF1    |  Emb-F1  |
> | :------------ | :--: | :------: | :------: | :--: | :------: | :------: | :--: | :------: | :------: |
> | Qwen2.5-VL-3B | 12.7 |   13.0   | 13.2 | 17.9 | 18.3 |   18.2   | 2.8  |   3.0    | 3.2  |
> | + SFT         | 51.3 |   52.9   | 53.3 | 49.8 |   52.0   | 52.4 | 22.4 | 23.0 |   22.8   |
> | Qwen2.5-VL-7B | 59.4 | 59.7 |   59.7   | 49.1 | 51.2 |   49.4   | 25.8 |   27.2   | 27.6 |
> | + SFT         | 70.9 |   71.9   | 72.3 | 55.4 |   57.7   | 58.2 | 29.5 |   31.4   | 31.5 |
> | InternVL3-2B  | 15.7 |   17.0   | 17.1 | 18.4 |   19.5   | 19.8 | 4.6  | 5.7  |   5.5    |
> | + SFT         | 34.6 |   39.4   | 39.9 | 37.8 |   40.7   | 41.2 | 13.7 |   14.7   | 15.0 |

---

> ### Author Response · Authors · 2025-11-23
> **Author response (7/7)**
>
> > **(Q3)** What is performance when training on one renderer and testing on another (full cross‑matrix), beyond the single‑renderer ablation in Table 5?
>
> Following the reviewer's request, **we added an additional experiment** to provide the following full cross-renderer evaluation results.
>
> As expected, each model achieves its best performance when trained and tested on the same renderer. However, even when the training data comes exclusively from one renderer, the model achieves substantially higher accuracy on unseen renderers than its base model, showing that the observed robustness reflects genuine gains in structural understanding.
>
> | Model                | Mermaid  | Graphviz | PlantUML | Diagrams  |
> | :------------------- | :------: | :------: | :------: | :-------: |
> | Qwen2.5-VL-3B        |   13.1   |   9.8    |   15.4   |    7.9    |
> | +SFT (Only Mermaid)  | **39.9** |   26.2   |   35.6   |   19.3    |
> | +SFT (Only Graphviz) |   32.8   | **40.3** |   31.9   |   23.2    |
> | +SFT (Only PlantUML) |   34.7   |   26.0   | **47.3** |   17.9    |
> | +SFT (Only Diagrams) |   32.7   |   27.2   |   34.2   | **31.2**  |
> | Qwen2.5-VL-7B        |   21.1   |   16.6   |   24.4   |   14.9    |
> | +SFT (Only Mermaid)  | **49.9** |   35.1   |   38.8   |   32.4    |
> | +SFT (Only Graphviz) |   37.0   | **46.2** |   35.4   |   33.5    |
> | +SFT (Only PlantUML) |   40.6   |   31.4   | **52.5** |   25.1    |
> | +SFT (Only Diagrams) |   34.8   |   29.2   |   32.2   | **37.5**  |
> | InternVL3-2B         |   11.8   |   8.5    |   12.2   |    5.0    |
> | +SFT (Only Mermaid)  | **34.2** |   21.1   |   31.3   |   15.0    |
> | +SFT (Only Graphviz) |   26.7   | **33.5** |   28.1   |   15.4    |
> | +SFT (Only PlantUML) |   27.6   |   21.0   | **42.7** |   11.3    |
> | +SFT (Only Diagrams) |   23.0   |   18.7   |   25.7   | **20.6**  |
> | MiniCPM-V2.6-8B      |   4.8    |   3.4    |   6.1    |    3.0    |
> | +SFT (Only Mermaid)  | **30.8** |   18.7   |   29.8   |   11.8    |
> | +SFT (Only Graphviz) |   20.8   | **29.2** |   23.6   |    9.7    |
> | +SFT (Only PlantUML) |   26.7   |   22.0   | **43.9** |   12.9    |
> | +SFT (Only Diagrams) |   23.2   |   18.6   |   26.7   | **22.67** |
>
> > **(Q4)** Can you quantify the gap between virtual‑node split/merge and a gateway/edge‑semantics encoding (e.g., BPMN‑style gateways) on hdBPMN?
>
> Our response to W2 has addressed this question. The results support our design choice.

---

> ### Author Response · Authors · 2025-11-27
> **A gentle reminder**
>
> Dear Reviewer,
>
> As the discussion period is drawing to a close, we wanted to kindly follow up on our responses. We sincerely hope they have addressed your comments satisfactorily, and we would be happy to provide any additional information or clarification if necessary.
>
> Thank you very much for your time and consideration.
>
> Best regards,
>
> The Authors

---

### Official Review · Reviewer_9ofn · 2025-10-28

**Soundness:** 3
**Presentation:** 3
**Contribution:** 4
**Rating:** 6
**Confidence:** 5

**Summary:**

The paper presents FlowGen, a controllable synthesizer that generates flowcharts with adjustable structural and visual complexity to evaluate and fine-tune multimodal large language models (MLLMs). The system defines parameters for graph order, branching, density, unlabeled edges, and nesting depth, and renders diagrams using multiple backends (Mermaid, Graphviz, PlantUML, Diagrams). The authors generate 11,520 flowcharts across 120 topic domains and three difficulty levels, fine-tune ten MLLMs, and evaluate on multiple flowchart parsing and QA datasets. Fine-tuning with FlowGen consistently improves performance, particularly on structurally complex diagrams. In addition, the authors use FlowGen to create challenging validation sets designed to probe model robustness along two controlled axes (graph complexity and scanned document degradation) which systematically expose consistent weaknesses in both open-source and proprietary models.

**Strengths:**

- **Novel and practical contribution:** FlowGen fills a significant gap in controllable synthetic flowchart generation for multimodal reasoning.
- **Controlled and systematic flowchart synthesis:** The modular three-stage architecture (configuration, graph construction, rendering) allows manipulation of structural and stylistic parameters, producing flowcharts with diverse properties.
- **Comprehensive experimentation:** Evaluations span multiple models, datasets, and difficulty levels, with well-designed ablations and detailed error analyses.
- **Robustness dimension:** The introduction of degraded and complex test sets is a valuable step toward stress-testing MLLMs.

**Weaknesses:**

- **Limits of flexibility and variability:** While FlowGen enables controlled chart variability, the actual range of possible flowcharts likely depends on the set of semantic topics and predefined parameter ranges. A quantitative study of generated flowchart diversity within the same topic would help clarify the boundaries and potential limitations of the method. Moreover, the 120 topic domains used for across the experiments are insufficiently described; the source and diversity of these domains need more explanation.
- **Unclear evaluation protocol:** The authors do not specify whether reported results on flowchart parsing and QA datasets are in zero-shot settings or if models were fine-tuned on the target evaluation datasets. This ambiguity makes it difficult to interpret the performance gains and assess true generalization.
- **Dataset independence concerns:** In section 4, training and test sets are both generated by FlowGen, raising questions about evaluation fairness and true generalization. Clarifying how overlap and near-duplicates are prevented is essential.

**Questions:**

**Questions for Authors**

1. Are the reported results on flowchart parsing and QA datasets obtained in zero-shot mode, or were the models fine-tuned on these evaluation datasets? Please clarify to properly interpret generalization performance.
2. When using FlowGen-generated training and test sets, how do the authors ensure no overlap or near-duplicates exist between them?
3. What is the source and diversity of the 120 topic domains used for generation? Are they drawn from existing ontologies?
4. Have the authors computed any quantitative metrics to measure variability within the same topic and across difficulty levels? This could help to assess the true flexibility of FlowGen and potential limitations in coverage.

---

> ### Author Response · Authors · 2025-11-20
> **Author response (1/2)**
>
> We sincerely thank you for taking the time to review our paper and for your valuable feedback. Below, we address the concerns you raised.
>
> > **(W1)** Limits of flexibility and variability: While FlowGen enables controlled chart variability, the actual range of possible flowcharts likely depends on the set of semantic topics and predefined parameter ranges. A quantitative study of generated flowchart diversity within the same topic would help clarify the boundaries and potential limitations of the method. Moreover, the 120 topic domains used for across the experiments are insufficiently described; the source and diversity of these domains need more explanation.
>
> Thank you for raising this point. We fully agree that diversity is crucial for robust model evaluation.
>
> We clarify that FlowGen produces flowcharts with significantly **greater structural and stylistic variety than existing datasets** (as shown in Table 1 in the paper). In addition to basic shape variations, it incorporates diversity in arrow branching, subgraph nesting, and rendering effects. This leads to broader coverage of real-world flowchart characteristics and provides essential supervisory signals when used as training data to enhance model performance (as shown in Section 3). In this regard, node names and edge labels (i.e., topics) have minimal impact. In fact, as indicated by our experimental results in response to Reviewer #i9ZK, the empirical differences are modest when using labels generated by different LLMs or by humans.
>
> That said, we agree that clarifying the boundaries of FlowGen’s flexibility is valuable. To ensure full transparency and reproducibility, we note that the complete list of 120 topic domains is already available in `/Synthesizer/my_dictionary.py` in our anonymous GitHub repository. The source and diversity of these topics will be clarified as part of our response to your Q3. Furthermore, in response to your Q4, **we conducted an additional experiment measuring pairwise graph edit distance between flowcharts, which confirms substantial variability even within the same topic.**
>
> > **(W2)** Unclear evaluation protocol: The authors do not specify whether reported results on flowchart parsing and QA datasets are in zero-shot settings or if models were fine-tuned on the target evaluation datasets. This ambiguity makes it difficult to interpret the performance gains and assess true generalization.
>
> Thank you for raising this important question. We apologize for any confusion caused and would like to clarify that all results reported in Section 3 reflect **zero-shot performance** (see prompts in Appendix D). Specifically, **the models were not fine-tuned on any target evaluation dataset**, but were only fine-tuned on our synthetic training data (as indicated by +SFT in Table 3). We will make this point more explicit in the paper to emphasize that the performance gains are attributable to our synthetic training set, rather than to in-context learning or dataset-specific tuning.
>
> > **(W3)** Dataset independence concerns: In section 4, training and test sets are both generated by FlowGen, raising questions about evaluation fairness and true generalization. Clarifying how overlap and near-duplicates are prevented is essential.
>
> We appreciate the opportunity to clarify this point, which appears to have been unclear in our original description. We would like to clarify that FlowGen-generated flowcharts are designed to be used **either** as training data (as in Section 3) **or** as test data (as in Section 4), **but not for both in the same experiment**, in order to prevent potential biases.
>
> In the specific experiment corresponding to Table 6, we intentionally used FlowGen-generated data for both training and testing in order to stress-test generalization capability. The results show that **even when models could potentially exploit biases in the data, they still struggled on our test set**. This further suggests that the difficulty of our benchmark stems from the complex topological structures---such as branching, nesting, and diverse layouts---which are not easily learnable through superficial patterns.
>
> We will incorporate this clarification in the paper.

---

> ### Author Response · Authors · 2025-11-20
> **Author response (2/2)**
>
> > **(Q1)** Are the reported results on flowchart parsing and QA datasets obtained in zero-shot mode, or were the models fine-tuned on these evaluation datasets? Please clarify to properly interpret generalization performance.
>
> **As detailed in our response to W2,** all reported results were obtained under a zero-shot evaluation setting. The models were not fine-tuned on any of the target evaluation datasets, ensuring that the performance gains reflect genuine generalization ability acquired through training on our synthetic data.
>
> > **(Q2)** When using FlowGen-generated training and test sets, how do the authors ensure no overlap or near-duplicates exist between them?
>
> **As detailed in our response to W3,** the flowcharts generated by FlowGen are designed to be used either as training data or as evaluation benchmarks, but not for both within the same experiment. The case of Table 6 is an exception, where we intentionally used FlowGen for both training and testing in a stress-test scenario. This setup was designed to demonstrate that even if potential biases or overlaps were exploitable, models still struggle with our benchmark, underscoring the inherent difficulty posed by its structural complexity.
>
> > **(Q3)** What is the source and diversity of the 120 topic domains used for generation? Are they drawn from existing ontologies?
>
> Thank you for this question. **We will incorporate the following explanation into the paper.**
>
> As outlined in our Prompt for Topic and Node/Edge Names Generation in Appendix D, we first used GPT-4o to produce a pool of candidate topics related to real-world flowchart scenarios. These candidates were manually filtered to remove inappropriate entries and consolidate similar ones. Throughout this process, we drew on a variety of publicly available sources---including industry documentation, course syllabi, and domain-specific datasets---to minimize redundancy, ensure terminological accuracy, and maintain broad domain diversity. This resulted in a final set of 120 distinct topics, each accompanied by its own vocabulary of node names and edge labels. The topics span diverse areas such as business operations, scientific processes, industrial engineering, software systems, healthcare, finance, education, and public services, ensuring wide coverage across multiple fields.
>
> Below are several sampled topics. The full list is available in `/Synthesizer/my_dictionary.py` in our anonymous GitHub repository.
>
> ```python
> ['Finance', 'Healthcare', 'Manufacturing', 'Education', 'E_commerce', 'Transportation', 'Energy', 'Legal', 'Logistics',  'Telecom', 'Agriculture','Tourism', 'Real Estate','Public Safety','Entertainment',  'Environment','Government', 'Retail', 'Automotive', 'Aerospace',  'Insurance',  'Cybersecurity', 'Smart Home', 'Smart City', 'Biotechnology', 'Pharmaceuticals', 'Robotics', 'Construction', 'Media', 'Sports', 'Fashion', 'Human Resources', 'Recruitment','Customer Service', 'Supply Chain', 'Mining','Maritime', 'Space Exploration', 'Climate Science', 'Meteorology','Waste Management', 'Water Management', 'Recycling',  'Forestry', 'Veterinary', 'Childcare',  'Elderly Care', 'Disaster Management', 'Military',   'Aviation', 'Navigation', '3D Printing', 'Nanotechnology', 'Quantum Computing',  'AR or VR', 'Gaming', 'Music',   'Film Production', 'Content Creation', ...]
> ```
>
> > **(Q4)** Have the authors computed any quantitative metrics to measure variability within the same topic and across difficulty levels? This could help to assess the true flexibility of FlowGen and potential limitations in coverage.
>
> Thank you for this constructive suggestion. **We have conducted an additional experiment to quantitatively evaluate the diversity of flowcharts generated under the same topic, which confirms significant variability.**
>
> Specifically, we computed the graph edit distance (GED) between pairs of flowcharts---that is, the minimum number of elementary graph operations (node/edge insertion, deletion, or substitution) required to transform one flowchart into another. The table below reports the mean GED with standard deviation (SD) between flowcharts from the same topic, both within the same difficulty level (top) and across different difficulty levels (bottom). The results reveal substantial dissimilarity among intra-topic flowcharts, with mean GED values ranging from 18.0 to 36.8 for those at the same difficulty level, and increasing to 76.8 for those across difficulty levels. These findings clearly demonstrate the considerable diversity of flowcharts generated even within a single topic.
>
> | | Mean GED | SD |
> | - | :--: | :--: |
> | **Intra-topic, same difficulty level** |      |      |
> | easy | 18.0 | 2.2  |
> | medium | 28.5 | 4.1  |
> | hard | 36.8 | 6.4  |
> | **Intra-topic, cross-difficulty** |      |      |
> | easy-medium | 38.0 | 5.1  |
> | easy-hard | 76.8 | 8.9  |
> | medium-hard | 48.5 | 7.2  |

---

> > ### Comment · Reviewer_9ofn · 2025-11-24
> >
> > I thank the authors for their detailed response and for addressing my suggestions. While the selection of topics could still be more clearly justified, I believe the paper now merits an Accept recommendation.

---

### Official Review · Reviewer_i9ZK · 2025-11-02

**Soundness:** 3
**Presentation:** 3
**Contribution:** 3
**Rating:** 6
**Confidence:** 4

**Summary:**

The paper introduces a system called FlowGen, which is a controllable flowchart synthesizer that generates diverse flowcharts with adjustable structural complexity and visual styles. The system enables large-scale, customizable datasets for both training and benchmarking multimodal large language models (MLLMs) on flowchart understanding and visual reasoning tasks. FlowGen supports multiple renderers (Mermaid, Graphviz, PlantUML, Diagrams) and configurable parameters (e.g., graph order, branching factor, nested subgraphs). Experimental results show that models trained with FlowGen data (e.g., Qwen2) achieve significant improvements in flowchart parsing and flowchart QA, with strong generalization to public datasets such as FlowVQA and FlowLearn. Moreover, FlowGen serves as a challenging evaluation benchmark, revealing weaknesses in even top proprietary models (GPT-4o, Gemini-2.5-Flash).

**Strengths:**

1. The motivation of the paper is strong. The authors motivate that prior work (including FlowchartQA, FlowVQA, FlowLearn) does not have control over structural and stylistic complexity, whereas FlowGen specifically introduces a parametric generation framework to address this limitation. The generation of synthetic data resembles the data augmentation which show indeed improvement in performance.

2. The paper presents a good experimental design with many MLLMs and two types of evaluation metrics (exact and relaxed metrics). The Easy/Medium/Hard complexity of the dataset seems interesting and allows for various analysis.

3. The paper contains good ablation studies that systematically isolate contributions from multiple perspectives: multi-renderer training; nested subgraphs; and split/merge arrows
showing their distinct impacts on performance.

**Weaknesses:**

1. Semantic labels (node and edge names) are generated with LLMs and verified by humans. I wonder if this could introduce stylistic or linguistic biases, especially if similar LLMs are later tested on FlowGen data.

2. FlowGen by design enables controllable generation, but it would be interesting to see if synthetic diagrams may or may not fully capture the diversity and imperfections of real-world flowcharts (e.g., hand-drawn figures, inconsistent layouts, textual clutter).

3. Since training and test distributions share the same generative pipeline, models might implicitly learn renderer-specific biases, even if content instances differ. Did the authors observed such issues?

**Questions:**

What could happen if the authors generate data with different LLMs (open-source vs. closed-source MLLMs)? What is the impact of a less performant MLLM? To what extent the noise hurts the overall performance?

Does a combination of MLLM generated and human generated data could improve performance and avoid potential biases that could be brought by the MLLM?

---

> ### Author Response · Authors · 2025-11-19
> **Author response (1/2)**
>
> We sincerely thank you for taking the time to review our paper and for your valuable feedback. Below, we address the concerns you raised.
>
> > **(W1)** Semantic labels (node and edge names) are generated with LLMs and verified by humans. I wonder if this could introduce stylistic or linguistic biases, especially if similar LLMs are later tested on FlowGen data.
>
> We appreciate this insightful comment. To address this point, **we conducted an additional experiment and found no significant evidence of LLM-specific biases.**
>
> Specifically, we regenerated all semantic labels using two alternative LLMs---Gemini-2.5-Flash and Llama-3-8B---in place of GPT-4o, which was originally used in our work. We then repeated our flowchart parsing experiment (Table 3 in the paper) using training sets synthesized with these LLMs. As shown in the F1 results below, models fine-tuned on any of the three synthesized datasets (+SFT) achieve consistently strong performance gains across different models and benchmarks. Moreover, the differences in performance among the three training sets are negligible, with no clear advantage for any single LLM. We will include the complete results of this experiment in the revised version of the paper.
>
> | **Model**               | **FlowVQA** | **CBD**  | **FC_A** | **FC_B** |
> | ----------------------- | ----------- | -------- | -------- | -------- |
> | **Qwen2.5-VL-3B**           | 12.7        | 17.9     | 2.8      | 8.6      |
> | +SFT (GPT-4o)           | 51.3        | 49.8     | 22.4     | **34.6** |
> | +SFT (Gemini-2.5-Flash) | **52.1**    | **50.4** | **22.9** | 34.1     |
> | +SFT (Llama-3-8B)       | 49.4        | 47.9     | 21.6     | 32.7     |
> | **Qwen2.5-VL-7B**       | 59.4        | 49.1     | 25.8     | 32.0     |
> | +SFT (GPT-4o)           | **70.9**    | **55.4** | 29.5     | **40.8** |
> | +SFT (Gemini-2.5-Flash) | 70.5        | 55.2     | **30.1** | 40.3     |
> | +SFT (Llama-3-8B)       | 69.2        | 54.0     | 28.7     | 39.5     |
> | **InternVL3-2B**        | 15.7        | 18.4     | 4.6      | 8.3      |
> | +SFT (GPT-4o)           | 34.6        | **37.8** | **13.7** | 21.6     |
> | +SFT (Gemini-2.5-Flash) | 34.2        | 37.4     | **13.7** | **22.0** |
> | +SFT (Llama-3-8B)       | **35.8**    | 36.5     | 13.3     | 21.2     |
> | **MiniCPM-V2.6-8B**     | 17.6        | 20.4     | 7.5      | 8.8      |
> | +SFT (GPT-4o)           | **44.0**    | 37.9     | 13.4     | 22.9     |
> | +SFT (Gemini-2.5-Flash) | 43.9        | **38.6** | **13.9** | **23.5** |
> | +SFT (Llama-3-8B)       | 42.3        | 36.1     | 12.9     | 21.8     |
>
> > **(W2)** FlowGen by design enables controllable generation, but it would be interesting to see if synthetic diagrams may or may not fully capture the diversity and imperfections of real-world flowcharts (e.g., hand-drawn figures, inconsistent layouts, textual clutter).
>
> We fully agree that diversity is crucial for robust model evaluation. Thank you for raising this point.
>
> 1. **Enhanced Structural and Stylistic Diversity:** FlowGen already produces diagrams with significantly greater structural and stylistic variety than existing datasets (as shown in Table 1 in the paper). In addition to basic shape variations, it incorporates diversity in arrow branching, subgraph nesting, and rendering effects---including scanned-document style renderings. This leads to broader coverage of real-world flowchart characteristics, including irregular layouts.
>
> 2. **Generalization to Hand-Drawn Styles:** Although FlowGen does not explicitly simulate hand-drawn flowcharts, models trained on our data show substantial improvements on benchmarks such as FC_A, FC_B, and hdBPMN, which contain hand-drawn samples (see Table 3 in the paper). This indicates that FlowGen-enhanced training improves robustness to real-world visual noise.
>
> 3. **Future Work:** We appreciate this valuable suggestion and plan to explicitly integrate hand-drawn stylistic elements and other real-world artifacts in future versions to further increase data diversity and realism.

---

> ### Author Response · Authors · 2025-11-19
> **Author response (2/2)**
>
> > **(W3)** Since training and test distributions share the same generative pipeline, models might implicitly learn renderer-specific biases, even if content instances differ. Did the authors observed such issues?
>
> Thank you for raising this important point. We agree that such renderer-specific biases can naturally exist when the training and test distributions share the same generative pipeline. However, **these biases are orthogonal to the main focus of our work.**
>
> We would like to clarify that FlowGen-synthesized flowcharts are intended to be used **either** as a training set (as in Section 3 in the paper) **or** as a test set (as in Section 4 in the paper), **but not simultaneously for both**, precisely to avoid such biases. In the experiment corresponding to Table 6 in the paper, we intentionally used FlowGen-generated data for both training and testing to stress-test the generalization challenge---**even when potential renderer-specific biases could be exploited, models still struggled on our test set**. This further indicates that the difficulty of our benchmark stems primarily from the complex topological structures (e.g., branching, nesting, and diverse layouts) rather than superficial rendering artifacts.
>
> We will include this clarification in the revised version of the paper.
>
> > **(Q1)** What could happen if the authors generate data with different LLMs (open-source vs. closed-source MLLMs)? What is the impact of a less performant MLLM? To what extent the noise hurts the overall performance?
>
> Thank you for raising this question. **Our response to W1 has already addressed this point.** The performance improvements remain substantial and consistent across training sets generated by different LLMs, including both open-source and closed-source MLLMs. In particular, the differences between using GPT-4o and a less performant MLLM such as Llama-3-8B are negligible, indicating that the noise introduced by a weaker LLM does not significantly impact overall performance.
>
> > **(Q2)** Does a combination of MLLM generated and human generated data could improve performance and avoid potential biases that could be brought by the MLLM?
>
> Thank you for this insightful question. Following your suggestion, **we conducted an additional experiment and obtained interesting findings.**
>
> Specifically, we manually curated the node names and edge labels for 20 out of the 120 topics originally generated by GPT-4o. As shown in the F1 results below, combining LLM-generated data with human-curated data indeed led to improved performance, as expected. However, the improvements---ranging from +0.2 to +1.2 points---were relatively modest. This suggests that the primary supervisory signals provided by FlowGen stem from its structural and stylistic diversity, rather than from the precise quality of semantic labels, which aligns with our original design intention. We will include this observation in the revised version of the paper.
>
> | **Model**                           | **FlowVQA** | **CBD**  | **FC_A** | **FC_B** |
> | ----------------------------------- | ----------- | -------- | -------- | -------- |
> | **Qwen2.5-VL-3B**                       | 12.7        | 17.9     | 2.8      | 8.6      |
> | +SFT (120 topics by GPT-4o)            | 51.3        | 49.8     | 22.4     | 34.6     |
> | +SFT (100 topics by GPT-4o + 20 topics by human) | **52.5**    | **50.6** | **23.3** | **35.4** |
> | **Qwen2.5-VL-7B**                   | 59.4        | 49.1     | 25.8     | 32.0     |
> | +SFT (120 topics by GPT-4o)            | 70.9        | 55.4     | 29.5     | 40.8     |
> | +SFT (100 topics by GPT-4o + 20 topics by human) | **72.0**    | **56.1** | **30.4** | **41.2** |
> | **InternVL3-2B**                    | 15.7        | 18.4     | 4.6      | 8.3      |
> | +SFT (120 topics by GPT-4o)            | 34.6        | 37.8     | 13.7     | 21.6     |
> | +SFT (100 topics by GPT-4o + 20 topics by human) | **35.5**    | **38.3** | **14.5** | **22.8** |
> | **MiniCPM-V2.6-8B**                 | 17.6        | 20.4     | 7.5      | 8.8      |
> | +SFT (120 topics by GPT-4o)            | 44.0        | 37.9     | 13.4     | 22.9     |
> | +SFT (100 topics by GPT-4o + 20 topics by human) | **44.2**    | **38.1** | **13.8** | **24.0** |

---

> ### Author Response · Authors · 2025-11-27
> **A gentle reminder**
>
> Dear Reviewer,
>
> As the discussion period is drawing to a close, we wanted to kindly follow up on our responses. We sincerely hope they have addressed your comments satisfactorily, and we would be happy to provide any additional information or clarification if necessary.
>
> Thank you very much for your time and consideration.
>
> Best regards,
>
> The Authors

---

### Meta-Review · Area_Chair_MMHX · 2026-01-06

**Summary:**

The paper introduces FlowGen, a controllable synthesizer for generating diverse flowchart datasets to train and benchmark MLLMs, which reviewers praised for its strong motivation and experimental breadth. While initial concerns focused on generalization to real-world data and the reliance on intermediate triplet extraction, the authors provided a robust rebuttal with extensive new experiments. These additions, particularly the end-to-end evaluations and cross-domain checks, successfully demonstrated the tool's utility for improving MLLM performance on both specific flowchart tasks and broader benchmarks.

**Reviewer Concerns:**

The rebuttal effectively resolved the primary disputes regarding evaluation protocols and semantic bias by conducting end-to-end fine-tuning experiments and demonstrating that structural learning occurs independently of label semantics. The authors also addressed generalization concerns by showing positive transfer to broader benchmarks and real-world hand-drawn datasets. The only remaining minor point involves the distinction between synthetic "scanned" augmentations and real document noise, but the empirical transfer results largely mitigate this concern regarding ecological validity.

**Reviewer Scores:**

The consensus shifted positively during the discussion phase, with Reviewer 9ofn explicitly upgrading their recommendation to Accept after clarifications on dataset diversity. Reviewer TxLZ, who initially requested broader benchmarking and end-to-end testing, would likely increase their score as the authors fully satisfied these substantial requests, while Reviewer i9ZK remained supportive.

---

### Decision · Program_Chairs · 2026-01-26

Accept (Poster)